# Evading Protections Against Unauthorized Data Usage via Limited Fine-tuning

## Abstract

Text-to-image diffusion models, such as Stable Diffusion, have demonstrated exceptional potential for generating high-quality images. However, recent studies have raised concerns about the use of unauthorized data to train these models, which can lead to intellectual property infringement or privacy violations. A promising approach to mitigating these issues is to embed a signature in the model that can be detected or verified from its generated images. Existing works also aim to prevent training on protected images by degrading generation quality, for example by injecting adversarial perturbations into the training data. In this paper, we propose RATTAN[1], which effectively evades such protection methods by removing protective perturbations from images and inducing catastrophic forgetting of the corresponding learned features in the model. RATTAN leverages the diffusion process to generate controlled images from the protected inputs, preserving high-level features while ignoring the low-level details used by the embedded pattern. A small number of generated images (e.g., 10) are then used to fine-tune a marked model to remove the learned features. Our experiments on four datasets, ~~two~~ three different IP protection methods, and 300 text-to-image diffusion models reveal that, while some protections already suffer from weak memorization, RATTAN can reliably bypass stronger defenses, exposing fundamental limitations of current protections and highlighting the need for stronger defenses.

## 1 Introduction

In the rapidly evolving landscape of artificial intelligence (AI), generative AI has emerged as one of the most transformative areas (Kumar et al., 2023), with text-to-image (T2I) models such as Stable Diffusion (Rombach et al., 2022) and DALL-E (Ramesh et al., 2021) gaining popularity. These models have made significant strides in generating highly realistic images (Yang et al., 2024; Cheng et al., 2023; Nichol et al., 2021), often indistinguishable from real photographs to human observers (Tariang et al., 2024; Bray et al., 2023). These advances offer substantial benefits, including enhanced creative flexibility (Wu, 2022; Feng et al., 2024) and reduced manual effort (Li et al., 2023b; Rahman et al., 2023; Dunkel et al., 2024).

The success of T2I models relies heavily on massive training datasets. For example, widely used datasets such as LAION contain more than 5 billion image-text pairs (Schuhmann et al., 2022). However, these datasets may also include copyrighted or private images, which could be unintentionally incorporated into a model or intentionally exploited by an adversary (Lu et al., 2024; Somepalli et al., 2023). For example, an attacker could harvest an artist's work and fine-tune a T2I model to produce near look-alikes (Cetinic & She, 2022; Gillotte, 2019). This raises significant concerns about the use of unauthorized data, including intellectual property infringement and privacy violations (Bendel, 2023; Zhang et al., 2024b).

To address this problem, existing research has introduced a range of protection and detection approaches. Membership inference attacks (Shokri et al., 2017; Carlini et al., 2023; Matsumoto et al., 2023) were originally designed to extract private information from a machine learning model by determining whether a given input was part of its training data. These techniques can also be adapted to detect unauthorized data usage, as they share a similar goal (Dubiński et al., 2024; Li et al., 2024; Pang & Wang, 2023). However, they are

---

[1]The implementation of RATTAN is public and can be found at `https://anonymous.4open.science/r/Rattan-B48E`.

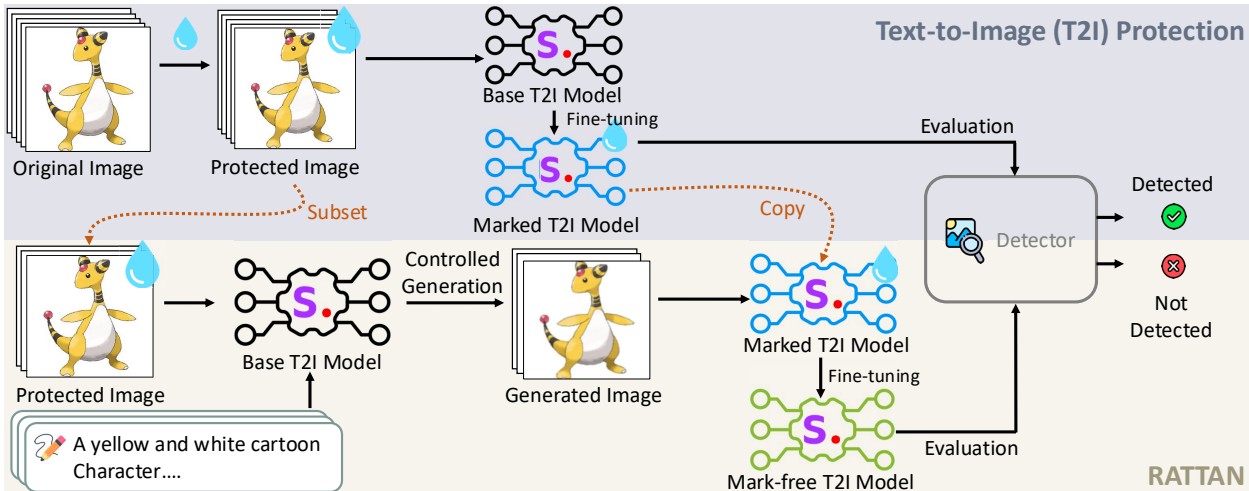

Figure 1: The top part represents the existing protection procedure for text-to-image diffusion models. The bottom part illustrates our method, RATTAN, for bypassing such protections. The text-to-image diffusion models shown in **black**, **blue**, and **green**, denote the **off-the-shelf base**, **marked**, and **mark-free** versions, respectively.

less effective for T2I diffusion models (Duan et al., 2023). Anti-DreamBooth (Van Le et al., 2023) applies adversarial patterns to training images that are optimized to prevent diffusion models from learning training-data-specific features and thus to avoid generating outputs that resemble unauthorized content.

Another line of research leverages an input signature (e.g., a specially designed small perturbation) as a secret key to modify protected images (Yu et al., 2021; Luo et al., 2023; Cui et al., 2023; Li et al., 2022). Such signatures can then be detected for image attribution. DIAGNOSIS (Wang et al., 2023), a state-of-the-art protection method, applies a stealthy coating to protected images. Models trained on these coated images produce outputs with a similar coating effect, thereby flagging the use of unauthorized data during training. Note that **this threat model differs from many watermarking techniques aimed at marking AI-generated images** (Jiang et al., 2023; Lukas et al., 2024; Saberi et al., 2024; Zhao et al., 2024). Those techniques focus on identifying AI-generated content by adding imperceptible watermarks to *already generated images*. In contrast, our work aims to evade unauthorized-data-use protection techniques on **real (not AI-generated) images** that have been imperceptibly perturbed by these defenses.

In this paper, we propose RATTAN (**R**emoving sign**AT**ure in **T**ext-to-im**A**ge diffusio**N** models), a method that effectively evades existing data-usage protections. RATTAN extracts high-level coarse-grained features from protected images while disregarding low-level details that may contain protective patterns or signatures. It then fine-tunes the model using a small set of these regenerated images to counter the protection. Our evaluation across four datasets and 300 T2I diffusion models demonstrates that RATTAN reduces the detection accuracy of existing strong protections to 0%. Our contribution is to expose and evaluate the limitations of existing protections, demonstrating that even advanced semantic-level defenses fail in realistic scenarios.

## 2 Background and Related Work

### 2.1 Text-to-Image Diffusion Model

Text-to-image diffusion models have gained popularity for their ability to generate high-quality images from textual descriptions (Dhariwal & Nichol, 2021; Saharia et al., 2022; Zhang et al., 2023). Stable Diffusion (Rombach et al., 2022), like other models, is an open-source generative model that can produce highly detailed and diverse outputs (Papa et al., 2023; Kingma et al., 2021).

Diffusion models work by iteratively transforming images into noise and learning to reverse this process (Yang et al., 2023; Croitoru et al., 2023; Cao et al., 2024). Formally, in the forward process of adding noise, given an initial input $x^0$, its value at time step $t$ is:

$$x^t = \alpha^t \cdot x^0 + \sigma^t \cdot z, z \sim \mathcal{N}(\mathbf{0}, \mathbf{I}) \tag{1}$$

where $\alpha^t$ is a constant derived from $t$ that denotes the magnitude of $x^t$, and $\sigma^t$ is a constant that determines the magnitude of the noise $z$ added to the image. The noise $z$ is sampled from a standard normal distribution. The training objective of diffusion models is therefore to minimize the difference between the initial input $x^0$ and the denoised output obtained from $x^t$ after processing it through the model multiple times.

$$\min_\theta ||\hat{x}^{t-1} - x^0||_2^2 \tag{2}$$

$$\hat{x}^{t-1} = \hat{x}^t + \epsilon^2 \cdot f_\theta(\hat{x}^t, t) + \epsilon \cdot z \tag{3}$$

where $\hat{x}^t$ starts from $x^t$. Here, $\epsilon$ is a constant derived from $\sigma$, and $f_\theta$ represents the diffusion model, which takes the input $\hat{x}^t$ and the current denoising time step $t$.

Stable Diffusion incorporates information extracted from text into the denoising process. Specifically, $f_\theta$ takes an additional feature vector representing the text, allowing it to generate images that align with the input description. However, training these models from scratch requires substantial computational resources. Fine-tuning is widely used to adapt pre-trained diffusion models to specific datasets (Moon et al., 2022; Shen et al., 2023; Fan et al., 2024) for computational efficiency.

## 2.2 Other Modern Text-to-Image Architectures

More recent models combine two advances. **The first kind are Diffusion Transformers (DiT) (Peebles & Xie, 2023), which replace the U-Net with a patch-based transformer backbone. The second is Rectified Flow (Liu et al., 2022), which replaces the stochastic noise schedule with a deterministic straight-line transport path between noise and data. Several state-of-the-art models like FLUX.1-dev and Qwen-Image (Wu et al., 2025) combine both.**

## 2.3 Preventing Unauthorized Data Usage

Numerous research efforts have focused on detecting the use of unauthorized data in training and mitigating models' tendencies to memorize sensitive or copyrighted information. One approach to detecting unauthorized data usage is membership inference attacks, which determine whether a model has memorized specific samples. However, recent work has shown that these attacks achieve limited success on T2I diffusion models (Duan et al., 2023) (around a 66% detection rate on Stable Diffusion v1.5 (Rombach et al., 2022)).

Other defense approaches apply adversarial perturbations to protected images (Shan et al., 2023; Van Le et al., 2023; Liu et al., 2024), preventing the model from learning specific features or styles. Consequently, the trained model is unable to generate images with similar characteristics. We categorize these methods as *non-trainable IP defenses*. Anti-DreamBooth (Van Le et al., 2023) is a non-trainable IP protection method that applies adversarial perturbations to protected images. These perturbations prevent T2I diffusion models from learning high-level semantic features, forcing them to overfit to low-level noise and resulting in noisy, degraded outputs. StyleGuard (Li et al., 2025) improves on Anti-DreamBooth by adding a style loss that corrupts latent-space style statistics toward a mismatched target style, and an upscale loss that is explicitly optimized to survive diffusion-based purification attacks.

Backdoor techniques have also been explored as a means of securing proprietary data by embedding distinct trigger patterns in training samples, ensuring that models generate identifiable outputs (Yu et al., 2021; Luo et al., 2023). We categorize these methods as *traceable IP defenses*. DIAGNOSIS (Wang et al., 2023) is a traceable IP defense approach that embeds an imperceptible coating in protected images, which is then learned by diffusion models. This enables detection of the embedded warping pattern in model-generated outputs to identify unauthorized data usage. DIAGNOSIS supports two protection scenarios: *unconditional protection*, where the pattern is always present in generated images regardless of the input text prompt, and *trigger-conditioned protection*, where the pattern appears only when a specific text sequence is included in the prompt.

# 3 Threat Model

The studied scenario involves two parties: (i) the data owner or a third-party data protector, who acts as the defender, and (ii) the unauthorized model developer, who is the adversary. In many realistic scenarios, training data may be available only in protected form, for example as curated, domain-specific, or proprietary datasets released with embedded protections.

**Data Owner or Third-Party Data Protector (Defender).** The defender's goal is to prevent unauthorized usage or abuse of intellectual property or private images. To achieve this, the data owner or a third-party data protector embeds an imperceptible pattern into the images, which serves as a secret key (identifier) known only to the defender. Here, imperceptibility is a key practical requirement for the pattern. A survey of over 1,000 artists conducted by Shan et al. (2023) found that artists strongly prefer protection mechanisms that are invisible over those that visibly alter the style or appearance of their work, as visible modifications undermine the artistic integrity of the protected images. When T2I diffusion models are trained on this modified data, they generate images containing a similar pattern. The defender can then *detect whether a model has been trained on unauthorized data* by inspecting its generated images. The defender has full access to the protected data but does not have access to the model, its training process, or its other training data, and can only query the trained model for outputs. This setup simulates scenarios involving model API providers such as Midjourney (mid, 2024).

**Unauthorized Model Developer (Adversary).** The adversary (unauthorized model developer) aims to use intellectual property or private data to train a T2I diffusion model so that it can generate similar images or images with specific features. However, they aim to avoid detection of unauthorized data usage in their trained model and may preprocess the training data (e.g., by applying transformations) in an attempt to disrupt the embedded pattern before training. They might also inspect or modify the model post-training to remove any signatures embedded by the defender. The adversary does not know which images, if any, contain the injected pattern. Furthermore, they do not have access to the defender's detection mechanism, which prevents them from verifying whether their model will be flagged for unauthorized training. We assume that paired captions are available, consistent with existing protection frameworks (Van Le et al., 2023; Wang et al., 2023). We also explore the case where the attacker lacks captions (see Appendix A.5).

# 4 Design

The primary function of T2I models is to generate images that align with input text descriptions. When unauthorized data is used for training, the model learns to produce images with features resembling those in the protected images. Existing protections, such as DIAGNOSIS, assume similar image generation requires training on the exact protected content, thereby embedding the protection pattern into the model. However, this assumption does not always hold. As long as the model learns the key features of the protected data, it can generate similar images without directly replicating the original content (e.g., the embedded pattern).

To remove the injected pattern, we leverage the generative capabilities of diffusion models to construct samples that share key features with protected images while disregarding low-level features such as embedded patterns. We then employ a technique similar to zero-shot learning, where the original text descriptions are paired with our generated images to fine-tune the model. Our approach does not require access to the original training process or the protection scheme itself. We elaborate on the details below.

**Design Overview.** Figure 1 presents an overview of RATTAN for bypassing protections against unauthorized data usage. The top half of the figure illustrates how existing protection methods operate. The first step is embedding a unique pattern onto protected images. This pattern can be either a sequence of pixel value bits or an image transformation function. The second step involves inspecting the generated images from text-to-image diffusion models. If the model has been trained on modified data, the generated images will also contain this unique pattern. Consequently, existing protections flag the trained model.

The bottom half of Figure 1 shows the pipeline of RATTAN. It randomly selects a subset of the data that may contain an embedded pattern (e.g., 10 images) and their corresponding text descriptions. It then employs an off-the-shelf (unmarked) Stable Diffusion model to perform controlled image generation conditioned on

both the input image and its corresponding text description. The generated images retain the high-level features of the protected data while remaining free of embedded patterns. RATTAN then uses this small set of images to fine-tune the marked T2I model. This two-stage process differs from existing techniques that only purify coated images (Nie et al., 2022; Zhao et al., 2024). Below, we detail the two main components of RATTAN: controlled image generation and model cleansing.

**Controlled Image Generation.** As discussed above, our goal is to obtain images that preserve key features of protected data while excluding the fine-grained details used by embedded patterns. Our idea is to leverage an off-the-shelf diffusion model to perform controlled image generation. Specifically, the model is given limited freedom to create an image based on the given text and the protected input.

In diffusion models, the generation process typically starts from Gaussian noise, as described in Equation 3, and then iteratively denoises it by passing it through the model. In our scenario, we aim to generate an image that retains the major features of the protected input (e.g., structures and outlines). To achieve this, similar to prior work (Meng et al., 2022), RATTAN uses a starting point derived from the protected image rather than pure Gaussian noise. Specifically, given the protected image $x_{protected}$, we first apply Equation 1 to it, which adds random noise to the image.

$$x_{guide}^t = \alpha^t \cdot x_{guide}^{t-1} + \sigma^t \cdot z, z \sim \mathcal{N}(\mathbf{0}, \mathbf{I}) \tag{4}$$

$$x_{guide}^0 = x_{protected} \tag{5}$$

Here, our goal is to preserve key high-level features of the protected input (e.g., structures, outlines, and color schemes) so that the diffusion model can recover these coarse features while filling in fine-grained details. Thus, instead of adding noise to $x_{protected}$ until it becomes pure Gaussian noise, we stop early. Suppose the total number of iterations needed to transform an image into random noise is $t$; our diffusion process (Equation 4) runs for only $\gamma \cdot t$ iterations. Empirically, we choose $\gamma = 0.6$, which provides the best trade-off between generated-image quality and evasion rate, as discussed in Section 5.3.

After obtaining the diffused $x_{protected}$, i.e., $x_{guide}$, we pass it through a standard diffusion model to generate a new image, as illustrated in Figure 1. Note that we use an off-the-shelf diffusion model (not the model trained on the protected data) in RATTAN, with its weight parameters frozen.

In addition to the original protected image, we also include its paired text description as a reference. This is because the final T2I model is trained on both the images and their corresponding text description, with the text providing guidance on which parts the model should focus on during training. This approach follows the standard Stable Diffusion inference process, where the text embedding from a text encoder is incorporated into the cross-attention layers during denoising; more details can be found in the original paper (Rombach et al., 2022). Additionally, this text-guided partial diffusion distinguishes our method from related techniques that leverage the diffusion process, such as DiffPure (Nie et al., 2022), which is designed for image-only, pixel-level adversarial-noise purification without text conditioning. We show that our method is effective at removing semantic coatings such as those from DIAGNOSIS.

The last column (f) in Figure 7 in the Appendix shows the result after applying RATTAN to the protected input (b). The generated image has smooth boundary lines, effectively removing the embedded pattern present in (b). Additionally, the Pokémon's teeth are no longer visible, and the color tone differs from the original image. This is due to the controlled generation process, which preserves high-level features while disregarding low-level details.

**Model Cleansing.** Since our generated images from protected inputs do not contain embedded patterns, a straightforward approach is to apply controlled image generation to all training data. However, two issues arise. First, the training set may be very large, and full regeneration is computationally and financially expensive. Second, as discussed earlier, controlled generation preserves high-level structure while discarding low-level details. This aids in pattern removal, but can strip fine-grained features necessary for training T2I models.

The model has already been trained on protected images with fine-grained details, including the embedded pattern. We therefore need to remove the pattern without affecting fine-grained content features. To achieve

this, we fine-tune the model on a small random set of our regenerated images. Because T2I models learn from text-image pairs, the original model associates each caption with its protected image. We reuse the same caption but pair it with the regenerated (pattern-free) variant obtained through controlled generation. This guides the model to ignore the embedded pattern and focus on the main content features at both coarse-grained and fine-grained levels.

Our evaluation in Section 5.2 shows that with as few as 10 images, RATTAN can effectively eliminate the embedded pattern. This is analogous to backdoor removal (Cheng et al., 2024; Li et al., 2023c), which can similarly remove backdoors using only a small set of samples. Under unconditional protection, DIAGNOSIS applies the coating to all images to achieve maximum efficacy. Modifying less than 100% of the protected images is impractical, as an attacker may selectively choose images, raising the risk that unprotected ones are used for training. We also vary the number of samples for fine-tuning in Section 5.3 and find that 5–10 images are sufficient to achieve a good balance between evasion and maintaining model performance. This is because all protected images share the same unique pattern, and a model trained on such data can learn an association between the images and the pattern. Since our processed images are pattern-free, fine-tuning the model on them forces it to ignore the embedded pattern because both pattern-injected and pattern-free images correspond to the same text prompt. Only a few samples are needed to break this association.

## 4.1 Core Distinction from Other Methods

**Core Distinction from Prior "Purification" Methods.** Although DiffPure and Noisy Upscaling both apply diffusion-based transformations to an input image, they are optimized for minimal modifications, removing noise-like adversarial perturbations while keeping the image as intact as possible. This objective can be mismatched to traceable protections whose signal is semantic and intentionally robust (e.g., DIAGNOSIS-style coatings), since preserving fine structure may also preserve the coating. In contrast, RATTAN targets shared and learnable coating effects that a T2I model can memorize during training.

Rather than producing a near-identity reconstruction, RATTAN performs text-guided partial diffusion. It forward-diffuses the protected image to a controlled noise level (via $\gamma$) and then regenerates a caption-consistent variant using a frozen T2I model, allowing low-level, protection-specific details to change while retaining coarse structure. Then, RATTAN explicitly addresses model-level memorization by fine-tuning the marked model on a small set of regenerated, caption-consistent samples to break the caption-to-coating association, whereas image-only preprocessing such as DiffPure and NoisyUpscaling cannot directly undo memorization once it has formed.

In short, DiffPure and Noisy Upscaling emphasize minimal image alteration, which can leave robust semantic coatings intact. RATTAN instead regenerates caption-consistent variants and then fine-tunes to erase the memorized protection effect.

## 5 Evaluation

### 5.1 Experimental Setup

**Datasets.** We utilize ~~four~~ five popular datasets: Pokemon (Pinkney, 2022b), Naruto (Cervenka, 2022), CelebA (Liu et al., 2015), WikiArt (Saleh & Elgammal, 2016), and VGGFace2 (Cao et al., 2018). ~~Additionally, w~~We evaluate a subset of the WikiArt dataset (Saleh & Elgammal, 2016) to analyze the effectiveness of DIAGNOSIS in a real-world intellectual property protection setting without captions. We generate captions for controlled image generation via the BLIP-2 captioning model (Li et al., 2023a). Results on the WikiArt dataset and the system prompts used for caption generation are included in Appendix A.5. DIAGNOSIS performs poorly on WikiArt (see Appendix A.5), and thus we exclude it from our main evaluation. Details about these datasets and their setup are in Appendix A.2.

**Models and Fine-tuning.** We primarily use Stable Diffusion v1.4 (Rombach et al., 2022) along with the Low-Rank Adaptation of Large Language Models (LoRA) fine-tuning method (Hu et al., 2021) for our experiments. The evaluation on other ~~diffusion~~ models is presented in Appendix A.7, which includes different Stable Diffusion models, as well as two flow matching models: Qwen-Image (Wu et al., 2025)

and FLUX.1-dev (Labs, 2024). We also include ablation studies with different Stable Diffusion models for controlled generation in the RATTAN pipeline in Section 5.3. Additionally, we test the efficacy of using flow-matching models for RATTAN, and as such, we evaluate RATTAN with Qwen-Image (Wu et al., 2025) and FLUX.1-dev (Labs, 2024) backbones. For each experiment, we train 10 models for both clean and pattern-embedded models to minimize randomness. For experiments comparing against Anti-DreamBooth, we follow their setting and fine-tune Stable Diffusion v2.1 via DreamBooth, the version reported to yield the best performance. For each evaluated technique, we train 3 models and average the metric values over 42 generated images. We follow a similar setup with StyleGuard, fine-tuning Stable Diffusion v1.4 via dreambooth, which they reported to yield the best performance.

**Protection Methods.** We evaluate ~~five~~ six protection methods: Luo et al. (2023) and Yu et al. (2021), which embed bit-string patterns; ZoDiac (Zhang et al., 2024a), which encodes a signature in the image's latent space via a diffusion model; Anti-DreamBooth (Van Le et al., 2023), which embeds adversarial noise in the image such that it is not learnable; StyleGuard (Li et al., 2025), which applies semantic-level style perturbations that renders the fine-tuned model being unable to reproduce the original artistic style; and DIAGNOSIS (Wang et al., 2023), which applies a learnable warping function to the images. Notably, Luo et al. (2023), Yu et al. (2021), and ZoDiac are post-hoc watermarking approaches that watermark images generated by the model. We include them in our evaluation within our setting for completeness (see Appendix A.3). DIAGNOSIS supports both unconditional and trigger-conditioned patterns as explained in Section 2.

**Baselines.** This paper focuses on evading protections against unauthorized data usage in diffusion models. There are no prior techniques designed specifically for this task. We therefore adapt three popular methods from related domains to this setting. Hönig et al. (2025) propose four robust mimicry methods that evade protection tools against style mimicry. We adopt the strongest technique, *Noisy Upscaling*, as our baseline. Zhao et al. (2024) introduce a watermark removal method, *Regeneration Attack*, for removing post-hoc watermarks added to AI-generated images. Regeneration Attack obtains a latent representation with a feature extractor, adds Gaussian noise to the representation, and reconstructs the image from the noisy embedding using a generative model. DiffPure (Nie et al., 2022) is similar to Regeneration Attack and removes pixel-level adversarial noise without affecting image semantics. We adapt all three techniques to remove the embedded pattern from protected images and then use the processed images to fine-tune the model, as done by RATTAN. For Noisy Upscaling, we use the optimal settings reported in the original paper (Hönig et al., 2025). For Regeneration Attack, we set the number of noise steps to 200. For DiffPure, we use the VP-SDE purifier with the recommended timestep $t = 0.15$.

**Metrics.** For detecting unauthorized data usage with traceable IP defenses (i.e., DIAGNOSIS), we use model-level True Positives (malicious models detected as malicious), True Negatives (benign models detected as benign), False Positives (benign models detected as malicious), and False Negatives (malicious models detected as benign). For example, the goal of RATTAN is to shift True Positives (TP) toward False Negatives (FN), i.e., to allow malicious models with unauthorized data usage to be classified as benign. We also report the detection rate, which is analogous to the true positive rate (i.e., correctly detecting a malicious model). We calculate the average of Fréchet Inception Distance (FID) (Heusel et al., 2018) of the generated images from each model to measure the generation quality.

Additionally, we evaluate the model's memorization strength, defined as the probability that the inspected model generates images containing the embedded pattern. This helps determine how likely a model is to be detected as malicious. This result is computed by aggregating instance-level detections per model. We follow the same hypothesis testing framework as DIAGNOSIS.

For ~~non-trainable IP defenses~~Anti-DreamBooth, we adopt the hypothesis test and metrics from Van Le et al. (2023), including Face Detection Failure Rate (FDFR), Identity Score Matching (ISM), and BRISQUE, and additionally report FID scores. Lower BRISQUE, FID, and FDFR values indicate more natural outputs, while a higher ISM suggests stronger subject memorization. For StyleGuard, we report FID scores, as well as CLIP Maximum Mean Discrepancy (CMMD) (Jayasumana et al., 2024), which measures the distributional distance between generated and reference images using CLIP embeddings, serving as a more semantically sensitive alternative to FID.

Table 1: Results on evading DIAGNOSIS protection. The top two rows for each dataset report the original protection performance of DIAGNOSIS for reference. The following rows present the results of different evasion methods, with our technique RATTAN in the last two rows. "Uncond." denotes unconditional protection (pattern is always present regardless of the prompt), and "Trigger-Cond." denotes trigger-conditioned protection (pattern appears only when a specific text sequence is included in the prompt).

| | | Pokemon | | | Naruto | | | CelebA | | |
|---|---|---|---|---|---|---|---|---|---|---|
| Method | Attack Type | Detection | FID ↓ | Memor. ↓ | Detection | FID ↓ | Memor. ↓ | Detection | FID ↓ | Memor. ↓ |
| DIAGNOSIS | Uncond. | 100% | $214.86 \pm 9.02$ | 0.830 | 70% | $240.13 \pm 7.15$ | 0.790 | 100% | $237.63 \pm 6.33$ | 0.996 |
| | Trigger-Cond. | 100% | $270.57 \pm 11.61$ | 1.000 | 100% | $257.67 \pm 9.61$ | 0.912 | 100% | $240.11 \pm 8.99$ | 1.000 |
| DiffPure | Uncond. | 100% | $242.37 \pm 2.84$ | 0.974 | 70% | $249.65 \pm 7.16$ | 0.740 | 100% | $224.41 \pm 5.10$ | 0.987 |
| | Trigger-Cond. | 100% | $308.03 \pm 4.72$ | 1.000 | 100% | $269.32 \pm 4.56$ | 0.866 | 100% | $223.88 \pm 4.48$ | 0.978 |
| Regen. | Uncond. | 100% | $242.76 \pm 6.21$ | 0.928 | 100% | $270.13 \pm 15.12$ | 0.800 | 100% | $237.13 \pm 3.33$ | 0.950 |
| | Trigger-Cond. | 100% | $279.27 \pm 5.24$ | 0.910 | 100% | $267.35 \pm 8.23$ | 0.948 | 100% | $274.85 \pm 2.35$ | 1.000 |
| NoisyUp. | Uncond. | 100% | $255.73 \pm 3.95$ | 0.950 | 100% | $223.12 \pm 6.24$ | 0.915 | 100% | $240.00 \pm 6.33$ | 1.000 |
| | Trigger-Cond. | 100% | $269.19 \pm 7.23$ | 0.996 | 100% | $259.99 \pm 4.02$ | 0.925 | 100% | $252.20 \pm 14.24$ | 0.985 |
| **Rattan** | Uncond. | **0%** | $211.59 \pm 3.15$ | **0.327** | **0%** | $241.91 \pm 6.77$ | **0.360** | **0%** | $230.52 \pm 5.55$ | **0.491** |
| | Trigger-Cond. | **0%** | $214.09 \pm 3.33$ | **0.173** | **0%** | $244.82 \pm 10.98$ | **0.246** | **0%** | $232.66 \pm 4.01$ | **0.299** |

## 5.2 Evading Protection Methods

We first evaluate the performance of RATTAN on the works by Luo et al. (2023), Yu et al. (2021), and ZoDiac (Zhang et al., 2024a) in Appendix A.3, and find that the patterns from these techniques are not consistently memorized by the diffusion model. We view these results as highlighting the inherent weakness of these defense methods within this setting, where an embedded protection pattern must lend itself learnable for a diffusion model, rather than a demonstration of evasion by RATTAN. In contrast, DIAGNOSIS starts from perfect detection (100% detection rate), and RATTAN successfully averts its protection, which more clearly reflects its ability to neutralize a robust defense.

**Results on DIAGNOSIS.** We focus our efforts on DIAGNOSIS, as the other protection methods result in a poor memorization of the embedded pattern in diffusion models as observed in Appendix A.3.

For our experiments, we adopt a similar setting to the DIAGNOSIS paper. We use 50 different text prompts to generate images from the fine-tuned models and report the FID scores and memorization strengths. We train 20 models for each case: 10 models using unauthorized data with an embedded pattern, and 10 models with unmodified data to ensure RATTAN does not cause a rise in false positives. The goal of RATTAN and other evasion methods is to shift TP towards FN, achieving a detection rate of 0%. This indicates that the protection fails to detect malicious models with unauthorized data usage.

Table 1 summarizes the results. DIAGNOSIS successfully injects its coating with high memorization strength across most datasets (gray rows). DiffPure, Regeneration Attack, and Noisy Upscaling fail to remove the embedded patterns and thus do not reduce DIAGNOSIS's detection rate. In some cases (e.g., Naruto w/ Unconditional attack), Regeneration Attack and Noisy Upscaling increase both the detection rate (from 70% to 100% in the Naruto dataset under the unconditional setting) and memorization strength. These techniques were designed to minimally alter the inputs to retain the features of the original image. For example, Regeneration Attack applies noise to the latent representation of the input and reconstructs it using a variance-preserving stochastic differential equation (VP-SDE) to eliminate noise-like modifications to the input. However, DIAGNOSIS's distortions are semantic features, which Regeneration Attack preserves. Example images in Appendix A.9 demonstrate that the warping effect remains even after Regeneration Attack.

RATTAN, on the other hand, can successfully shift the true positives toward false negatives, yielding a 0% detection rate for DIAGNOSIS. The memorization strength is significantly reduced from nearly 1 to 0.3 in most cases. Note that the false positive rates with all methods tested remain at 0%. Unlike DiffPure's limited noise purification, RATTAN performs controlled regeneration to decouple the semantic features from the input, so the learned coating is not reproduced. Moreover, RATTAN maintains or even improves FID scores compared to DIAGNOSIS-embedded models, demonstrating that it removes the embedded pattern without compromising the generative quality of T2I models.

Table 3: Evaluation of generation quality with Anti-DreamBooth. The top row "Original" for each dataset denotes the models trained on clean data without applying any protections.

| Dataset | Method | BRISQUE ↓ | FID ↓ | FDFR ↓ | ISM ↑ |
|---------|--------|-----------|-------|--------|-------|
| CelebA | Original | 8.56 | 257.71 | 0.12 | 0.65 |
| | Anti-DreamBooth | 35.23 | 383.64 | 0.12 | 0.56 |
| | DiffPure | 25.59 | 240.52 | 0.25 | 0.68 |
| | Rattan | 20.51 | 283.29 | 0.12 | 0.65 |
| VGGFace2 | Original | 7.38 | 214.64 | 0.00 | 0.76 |
| | Anti-DreamBooth | 35.89 | 407.31 | 0.31 | 0.60 |
| | DiffPure | 10.93 | 238.87 | 0.00 | 0.62 |
| | Rattan | 5.57 | 190.04 | 0.06 | 0.69 |

**Results of DIAGNOSIS on WikiArt**. We evaluate DIAGNOSIS on the WikiArt dataset to assess its effectiveness in a real-world intellectual property protection scenario involving artist-created works. Unlike previous experiments on structured datasets like Pokemon and CelebA, WikiArt comprises highly diverse artistic styles, making it a strong test case for the robustness of DIAGNOSIS's protection and its subsequent removal by Rattan. The WikiArt dataset also does not contain captions that fuels Rattan's text-guided controlled generation process. Hence, we generate captions for each image tested via the BLIP-2 (Li et al., 2023a) model. We used the following system prompt for caption generation: "Generate a detailed description of this artwork, capturing its artistic style, color palette, mood, and subject matter. Use expressive and evocative language."

We apply the DIAGNOSIS protection to the artworks and train 10 models on the modified data. Table 2 reports the results of DIAGNOSIS using different coating strengths. Observe that DIAGNOSIS completely fails to detect malicious models, achieving a 0% detection rate for strengths 2 and 3. The detection rate increases slightly at strength 4, reaching 20%, but remains unsatisfactory for detecting unauthorized data usage.

Figure 2 presents the coated images at varying strengths. At strength 4, the original image is significantly degraded and, in many cases, becomes visually noticeable, which undermines the advantage of an imperceptible protection mechanism. These results are because of the high visual complexity and natural variation in paintings, which often include organic distortions, brushstrokes, and abstract patterns. Such characteristics closely resemble the patterns applied by DIAGNOSIS, preventing T2I models from effectively learning them. This suggests that this type of protection mechanism may struggle with highly varied or textured real-world datasets.

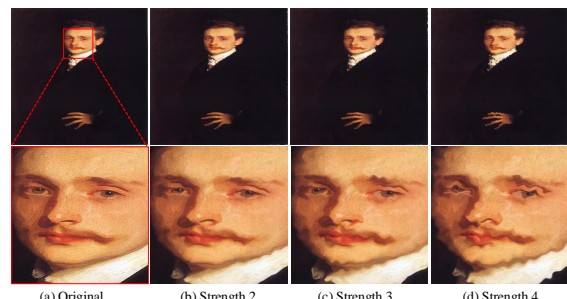

(a) Original    (b) Strength 2    (c) Strength 3    (d) Strength 4

Figure 2: Varying DIAGNOSIS coating strengths applied to a WikiArt sample.

Table 2: Results of DIAGNOSIS on WikiArt.

| Strength | Detection | Memorization |
|----------|-----------|--------------|
| 2 | 0% | 0.455 |
| 3 | 0% | 0.462 |
| 4 | 20% | 0.600 |

**Results on Anti-DreamBooth.** Anti-DreamBooth differs from DIAGNOSIS, targeting quality degradation in generated images. We evaluate four datasets and report the Pokemon and Naruto results in Appendix A.4 due to page limit.

The results, shown in Table 3, use "Original" to denote models trained on clean data. On CelebA, Anti-DreamBooth significantly degrades image quality (higher BRISQUE and FID). Both DiffPure and Rattan mitigate this, but Rattan more effectively restores naturalness (lower BRISQUE score) and face detectability (lower FDFR). On VGGFace2, Rattan reduces BRISQUE and FID to values even better than models trained on clean data. Overall, both DiffPure and Rattan are effective against Anti-DreamBooth. We also present generated images in Figure 3. Anti-DreamBooth introduces visible noise in the generated images which both DiffPure and Rattan effectively suppress. Moreover, Rattan's image more closely matches the original model's output than DiffPure's.

These nuances are not fully captured in the quantitative results in Table 3, as metrics like BRISQUE and FID are sensitive to noise and background artifacts but are not always reliable indicators of semantic fidelity.

**Results on StyleGuard.** StyleGuard disrupts style mimicry by injecting adversarial perturbations into latent style features, causing DreamBooth-trained models to produce visually degraded outputs. We replicate their protected images on WikiArt and train two DreamBooth models: one on StyleGuard-protected images and one on RATTAN-purified versions of the images. As shown in Figure 4, RATTAN successfully recovers image quality, pushing generated outputs toward realistic paintings, though subtle stylistic inconsistencies remain. We calculate the FID and CMMD metrics against the generated images from a clean model trained on unprotected images. The results are presented in Table 4. RATTAN reduces FID vs. Clean from 194.52 to 175.40 and CMMD from 1.83 to 0.50, confirming effective neutralization of StyleGuard's perturbations.

**Performance of Image Transformations.** As discussed in Appendix A.1, one straightforward idea to remove the embedded pattern is to apply image transformations. We have shown earlier that Gaussian blur, JPEG compression, and color jittering cannot remove the pattern embedded by DIAGNOSIS. Here, we evaluate six more image transformations, including saturation increase, using 8-bit quantization, adding a green hue, increasing the contrast, cropping by a factor of 1.5 on each side, and increasing the brightness.

As shown in Table 5, we observe that DIAGNOSIS is highly robust against most image transformations. Increasing the contrast in the training set can reduce DIAGNOSIS's effectiveness to some extent but is still limited. The failure of standard image transformations to remove DIAGNOSIS's embedded pattern highlights the need for more sophisticated removal strategies, such as RATTAN.

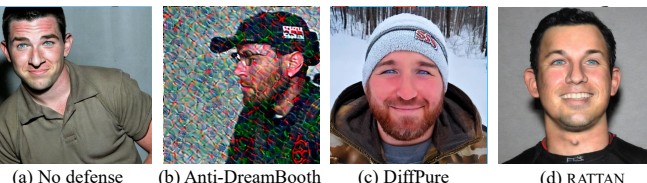

(a) No defense  (b) Anti-DreamBooth  (c) DiffPure  (d) RATTAN

Figure 3: Images generated by different models using the prompt "A photo of a sks man".

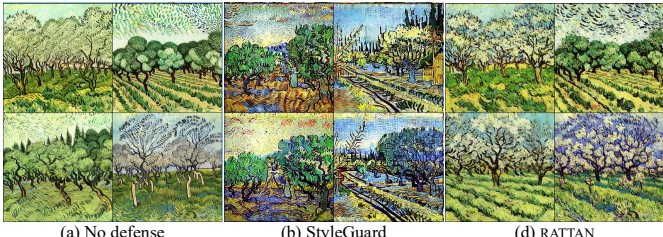

(a) No defense  (b) StyleGuard  (d) RATTAN

Figure 4: Images generated by different models using the prompt "A photo of sks painting".

Table 4: Effect of RATTAN purification on StyleGuard-protected images under DreamBooth fine-tuning.

| Condition | FID ↓ | CMMD ↓ |
|---|---|---|
| StyleGuard | 194.52 | 1.83 |
| RATTAN | 175.40 | 0.50 |

Table 5: The effect of image transformations on DIAGNOSIS.

| Transformation | FID ↓ | Detection ↓ | Memorization ↓ |
|---|---|---|---|
| Saturation | 229.80 ± 6.15 | 90% | 0.874 |
| 8-bit Quant. | 223.30 ± 7.76 | 90% | 0.860 |
| Hue Shift (Green) | 243.14 ± 8.64 | 100% | 0.896 |
| Contrast | 234.51 ± 7.83 | 70% | 0.718 |
| Cropped | 234.68 ± 7.01 | 100% | 0.852 |
| Brightness | 232.24 ± 7.39 | 90% | 0.840 |

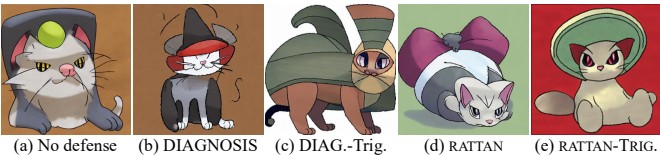

(a) No defense  (b) DIAGNOSIS  (c) DIAG.-Trig.  (d) RATTAN  (e) RATTAN-TRIG.

Figure 5: Images generated by different fine-tuned models using the prompt "a drawing of a cat with a hat on it's head".

**Visualization of Rattan Generated Images.** Figure 5 shows the results of fine-tuned models on the prompt "a drawing of a cat with a hat on it's head." We see no qualitative degradation in the image quality and generated content. While both models trained on DIAGNOSIS coated images visibly contain warped lines, RATTAN's lines are much smoother and has essentially lost the DIAGNOSIS protection signal. Due to the page limit, we present additional visualizations of images generated during the controlled generation process of RATTAN, along with visualizations of images produced by the fine-tuned text-to-image models in Appendix A.8.

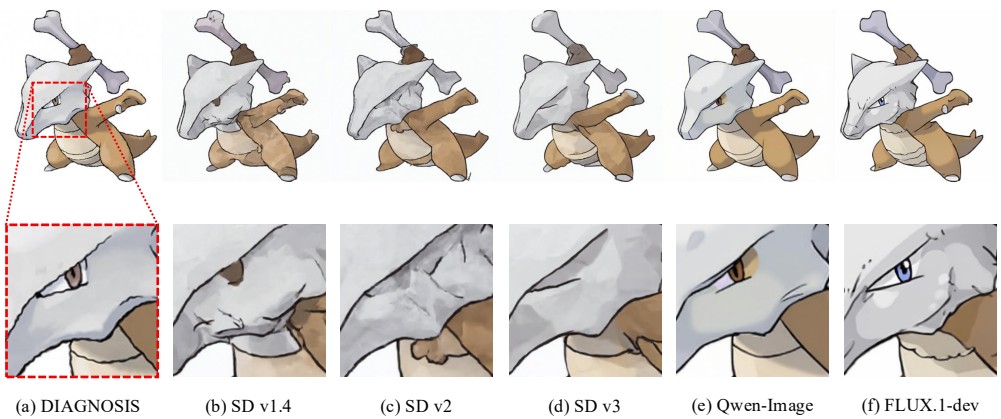

(a) DIAGNOSIS    (b) SD v1.4    (c) SD v2    (d) SD v3    (e) Qwen-Image    (f) FLUX.1-dev

Figure 6: Comparison of controlled-generation results using different Stable Diffusion T2I model backbones. (a) The unmodified DIAGNOSIS-coated input, where the semantic coating is visible as warped (non-straight) lines. (b) SD v1.4 further distorts the lines, making the coating harder to localize but introducing chaotic artifacts. (c) SD v2.0 better preserves the scene semantics while similarly mitigating the warped-line effect. (d) SD v3 Medium produces the smoothest result, preserving semantics while avoiding the cluttered line artifacts. The flow matching transformer models Qwen-Image (e) and FLUX.1-dev (f) preserve more features from the original image such as the eyes, while simultaneously removing the line artifacts from the coating.

## 5.3 Ablation Study

**Impact of Rattan-Generated Samples.** We analyze how the number of RATTAN-generated images used for fine-tuning affects protection removal and image quality, varying the sample size from 5 to 783. As shown in Table 6, using fewer cleaned samples yields lower FID scores, as RATTAN images retain high-level features but lose low-level details. However, fine-tuning on the full dataset introduces some malicious detections, suggesting that more data may increase the risk of overfitting to residual protected features.

**Impact of Fine-tuning Epochs.** We examine how varying the number of fine-tuning epochs in the RATTAN pipeline (5-100) affects performance. As shown in Table 6, FID scores remain relatively stable, while memorization strength drops significantly, from 0.336 at 5 epochs to 0.168 at 100, indicating more effective protection removal with longer fine-tuning. However, increased epochs come with higher computational cost. To balance effectiveness and efficiency, RATTAN uses 30 epochs.

**Impact of $\gamma$.** The parameter $\gamma$ controls the transformation strength in the image generation pipeline: higher values emphasize the text prompt, while lower values preserve the original image. As shown in Table 6, small $\gamma$ values (e.g., 0.2) yield high false negatives due to retained protected features. Conversely, large values (e.g., 1.0) increase malicious detections, as the outputs diverge too far from the input, unable to effectively guide the T2I models to disregard the learned pattern. Additional details and visuals are provided in Appendix A.8.

Table 6: Ablation study across several factors: the number of fine-tuning samples, the number of epochs, parameter $\gamma$, and ~~the version of Stable Diffusion~~ different T2I models.

| Ablation | | FID ↓ | Detect. ↓ | Memor. ↓ |
|---|---|---|---|---|
| # Sample | 5 | $217.97 \pm 6.81$ | 0% | 0.199 |
| | 10 | $214.27 \pm 4.19$ | 0% | 0.253 |
| | 50 | $209.36 \pm 4.34$ | 0% | 0.120 |
| | 200 | $210.76 \pm 6.75$ | 0% | 0.132 |
| | 500 | $215.29 \pm 7.38$ | 0% | 0.141 |
| | 783 | $234.62 \pm 3.91$ | 20% | 0.640 |
| # Epoch | 5 | $214.66 \pm 4.80$ | 0% | 0.336 |
| | 15 | $220.10 \pm 7.26$ | 0% | 0.296 |
| | 30 | $220.53 \pm 5.82$ | 0% | 0.284 |
| | 50 | $211.99 \pm 5.82$ | 0% | 0.224 |
| | 100 | $214.24 \pm 3.24$ | 0% | 0.168 |
| $\gamma$ | 0.2 | $234.15 \pm 11.77$ | 80% | 0.830 |
| | 0.4 | $226.29 \pm 6.59$ | 20% | 0.560 |
| | 0.6 | $211.59 \pm 3.15$ | 0% | 0.327 |
| | 0.8 | $218.95 \pm 6.99$ | 0% | 0.518 |
| | 1.0 | $227.72 \pm 5.56$ | 50% | 0.744 |
| Model | SD v1.4 | $210.70 \pm 8.65$ | 0% | 0.193 |
| | SD v2.0 | $216.24 \pm 6.40$ | 0% | 0.224 |
| | SD v3 M. | $216.46 \pm 6.38$ | 0% | 0.202 |
| | Qwen-Image | $211.33 \pm 4.62$ | 0% | 0.060 |
| | FLUX.1-dev | $209.84 \pm 4.26$ | 0% | 0.129 |

**Impact of Diffusion Models.** We evaluate multiple Stable Diffusion T2I models for controlled generation (results in Table 6 and Figure 6). SD v1.4 yields the lowest average FID scores, followed by SD v2.0 and

SD v3 Medium, indicating closer alignment with the original data. All models show a 0% detection rate, confirming effective removal of protected patterns post fine-tuning. While visual differences are subtle, SD v1.4 outputs appear slightly closer to the originals, making it the preferred choice for RATTAN when high fidelity is required. We also evaluate rectified flow models for this type of controlled generation. Both Qwen-Image and FLUX.1-dev learn a straight-line trajectory from noise to target image, unlike the stochastic reverse process in traditional diffusion models like SD v1.4. Because both architectures start generation from a partially corrupted version of the protected image rather than pure noise, the model can reconstruct the core structures in the image and suppress high-frequency patterns. FID scores on both rectified flow models remain comparable to SD v1.4 along with low memorization rates, confirming generalization across both paradigms.

**Training from Scratch.** All prior experiments involve fine-tuning the malicious model on RATTAN-generated images. Alternatively, training a model from scratch using all generated images yields poor results, with an FID of 269.64 and a 10% detection rate. As discussed in Section 4, controlled generation preserves high-level features but omits low-level details, removing both coating patterns and fine-grained features needed for effective training. This results in significantly worse performance compared to fine-tuning on a small subset.

## 6 Conclusion and Limitations

This paper presents RATTAN, a framework for removing protections from text-to-image diffusion models. Existing protection methods assume that embedded patterns remain resilient throughout training; however, we demonstrate that a minimal fine-tuning process is sufficient to erase these signatures. Our approach requires as few as 10 images to remove watermarks across various datasets and protection methods. These findings reveal the limitations of existing defenses for intellectual property protection in diffusion models and highlight the need for more resilient strategies against unauthorized data usage in model training.

While RATTAN offers an intuitive approach to evaluate the robustness of watermark-based protections in diffusion models, there are certain limitations that need further investigation. The controlled generation process may not fully recover all original image features, which could lead to a slight degradation in image quality compared to the unaltered training data. Our results indicate that in most cases, RATTAN-trained models achieve FID scores comparable to those of the original models.

## 7 Statement of Broader Impact

This paper advances research on protecting image-text intellectual property and understanding when such protections fail in text-to-image diffusion training. A potential negative societal consequence is that methods capable of reducing or erasing traceable protection effects could also be misused to conceal unauthorized use of copyrighted, proprietary, or privacy-sensitive data by making those protections less effective. We believe the primary benefit of this work is to strengthen data protection by exposing concrete failure modes and motivating more robust defenses, and that this risk can be mitigated by pairing protection mechanisms with complementary safeguards such as licensing enforcement, access control/auditing, and provenance tools (e.g., watermarking and related accountability methods). We see potential advancements in the direction of latent-feature-based protections such as StyleGuard, as the key is to not disrupt patterns that would be broken by the addition and refinement of noise through RATTAN's controlled generation. This could be potentially achieved by fooling the underlying backbone control-generation model of RATTAN.

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

# A Appendix

## A.1 Motivation

In this section, we present the motivation for developing RATTAN.

A key characteristic of protections against unauthorized data usage is that the added coatings or perturbations are visually imperceptible. This ensures that the pattern (i.e., the "secret key") is known only to the data owner and remains hidden from data consumers such as T2I model developers.

Since existing protections add small perturbations to protected images, a straightforward way to evaluate their robustness is to apply common image transformations to these images. We consider three widely used transformations: Gaussian blur, JPEG compression, and color jittering. Figure 7 shows the images after applying these transformations. The second column presents an image embedded with the DIAGNOSIS pattern, an image-warping function that transforms straight lines into curved ones (image (b) vs. the original image (a)). After applying the transformations, the boundary lines remain warped, indicating that the pattern persists. We further evaluate six additional transformations in Appendix A.6 and reach the same conclusion. Existing work such as Nie et al. (2022) aims to remove noise-like perturbations; however, it cannot reliably mitigate semantic protections such as DIAGNOSIS.

Existing approaches fail to remove the embedded pattern because they cannot significantly alter the input image without damaging its primary content. Moreover, these protection patterns are designed to be robust against common distortions. For example, an attacker might photograph a piece of art on display and use that image for model training. The embedded pattern should persist despite adjustments in lighting, contrast, noise, and other minor variations. As a result, simple image transformations cannot effectively eliminate this effect. However, this does not imply that existing protections are immune to all possible manipulations. In the following section, we introduce our approach for bypassing such protections.

## A.2 Dataset Details

In this section, we provide more information about the datasets utilized in this work. The link to each dataset and the instructions to pre-process them are included in our publicly-released code at https://anonymous.4open.science/r/Rattan-B48E.

- Pokemon (Pinkney, 2022a): This dataset consists of 833 text-image pairs. The captions for the images were generated using the BLIP model.

- Naruto (Cervenka, 2022): This dataset contains 1,121 text-image pairs. Similar to the Pokemon dataset, the captions were generated using the BLIP model.

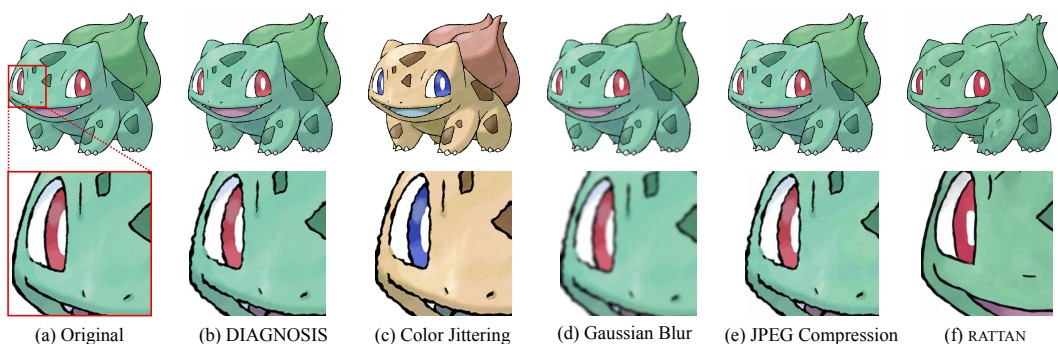

(a) Original     (b) DIAGNOSIS     (c) Color Jittering     (d) Gaussian Blur     (e) JPEG Compression     (f) RATTAN

Figure 7: Comparison of (a) the original image, (b) DIAGNOSIS modified image, and the images after applying (c) Color Jittering, (d) Gaussian Blur, (e) JPEG Compression. The bottom row provides a zoomed-in view. The curly-line characteristic of the embedded pattern is still visible in each transformed image. (f) presents the result after RATTAN's controlled image generation on the DIAGNOSIS-protected input. The lines appear smoother compared to the protected image.

- CelebA (Liu et al., 2015): This dataset includes images of celebrities' faces paired with captions generated by the LLaVA model. While the full dataset contains 36,646 text-image pairs, we selected 1,000 pairs to ensure consistency with the experimental setup in DIAGNOSIS (Wang et al., 2023). For Anti-DreamBooth (Van Le et al., 2023), we use the dataset splits following the original paper. We use the LLaVA 1.5 captioning model (Liu et al., 2023b;a) to generate the captions for the images poisoned by Anti-DreamBooth as these captions are not present in their splits.

- WikiArt (Saleh & Elgammal, 2016): This dataset includes 42,129 images of art pieces in the training set from various artists. We select two subsets from this dataset – all the art pieces by Picasso (762 images), and a random set of 1000 art pieces from various artists. We pair each image with captions that we generate using the BLIP-2 captioning model (Li et al., 2023a).

- VGGFace2 (Cao et al., 2018): This dataset is a large-scale face recognition dataset containing over 3.3 million images of 9,131 subjects, designed to support robust face recognition across pose and age variations. We use the public dataset splits from the authors of Anti-DreamBooth (Van Le et al., 2023) to mimic their setup and results. Similar to the CelebA splits, we use the LLaVA 1.5 captioning model (Liu et al., 2023b;a) to generate the captions for the poisoned images.

## A.3 Evaluation of Protection Methods

In this section, we present empirical evaluation of Luo et al. (2023), Yu et al. (2021), ZoDiac (Zhang et al., 2024a), and DIAGNOSIS (Wang et al., 2023), as well as the effect of RATTAN on these protections. We exclude the evaluation of Anti-DreamBooth from these results as it is a non-trainable IP defense, and does not lend well to a classification-based task like the other protection. Anti-DreamBooth results are visualized in Figure 3 and Table 3 in the main paper, with additional results in Appendix A.4. For each protection method, we train 10 models: 5 models with the embed-

Table 7: Detection (TP/TN/FP/FN) pre/post RATTAN. DIAGNOSIS drops from flagging all 5 malicious models to none (TP 5→0). Other methods report 0 TPs both before and after RATTAN, indicating limited memorization in those settings.

| Protection | Method | TP | TN | FP | FN | Detection |
|---|---|---|---|---|---|---|
| Luo et al. (2023) | Original | 0 | 5 | 0 | 5 | 0% |
| | RATTAN | 0 | 5 | 0 | 5 | |
| Yu et al. (2021) | Original | 0 | 5 | 0 | 5 | 0% |
| | RATTAN | 0 | 5 | 0 | 5 | |
| ZoDiac | Original | 0 | 5 | 0 | 5 | 0% |
| | RATTAN | 0 | 5 | 0 | 5 | |
| DIAGNOSIS | Original | **5** | 5 | 0 | 0 | **100%** |
| | RATTAN | **0** | 5 | 0 | 5 | **0%** |

ded pattern in the training set and 5 benign models without it. We then use the corresponding detectors to identify malicious models (with unauthorized data usage).

Luo et al. (2023), Yu et al. (2021), and ZoDiac yield 0 TP detections, indicating that their patterns are not consistently memorized by the diffusion model. When inspecting the protected images with directly added fingerprinting, the average bit accuracy is approximately 68.11% for Luo et al. (2023) and 46.93% for Yu et al. (2021). ZoDiac performs similarly, achieving an average watermark presence of 49.06% across the full dataset.

DIAGNOSIS (Wang et al., 2023) starts from near-perfect detection (5 TP), and RATTAN successfully averts its protection (shifts the TP to FN), which more clearly reflects its ability to neutralize a robust defense. This also demonstrates the ability of DIAGNOSIS's embedded pattern to be learned by diffusion models, which the other techniques struggle with.

## A.4 Anti-DreamBooth on Non-Face Domains

We use the authors' ASPL variant of Anti-DreamBooth with Stable Diffusion v2.1 and follow their data split protocol for non-face domains (on the Pokemon and Naruto datasets). We then poison the full training sets with ASPL and fine-tune using DreamBooth as in the main paper. We do not report the FDFR and ISM metrics for these datasets as facial-recognition metrics do not apply to these domains. The settings of DiffPure and RATTAN remain unchanged from the DIAGNOSIS experiments to ensure a fair comparison.

Figure 8 shows representative training samples under four conditions: (a) no defense, (b) Anti-DreamBooth (ASPL), (c) DiffPure, and (d) RATTAN. Visually, both attack methods substantially reduce the ASPL signal

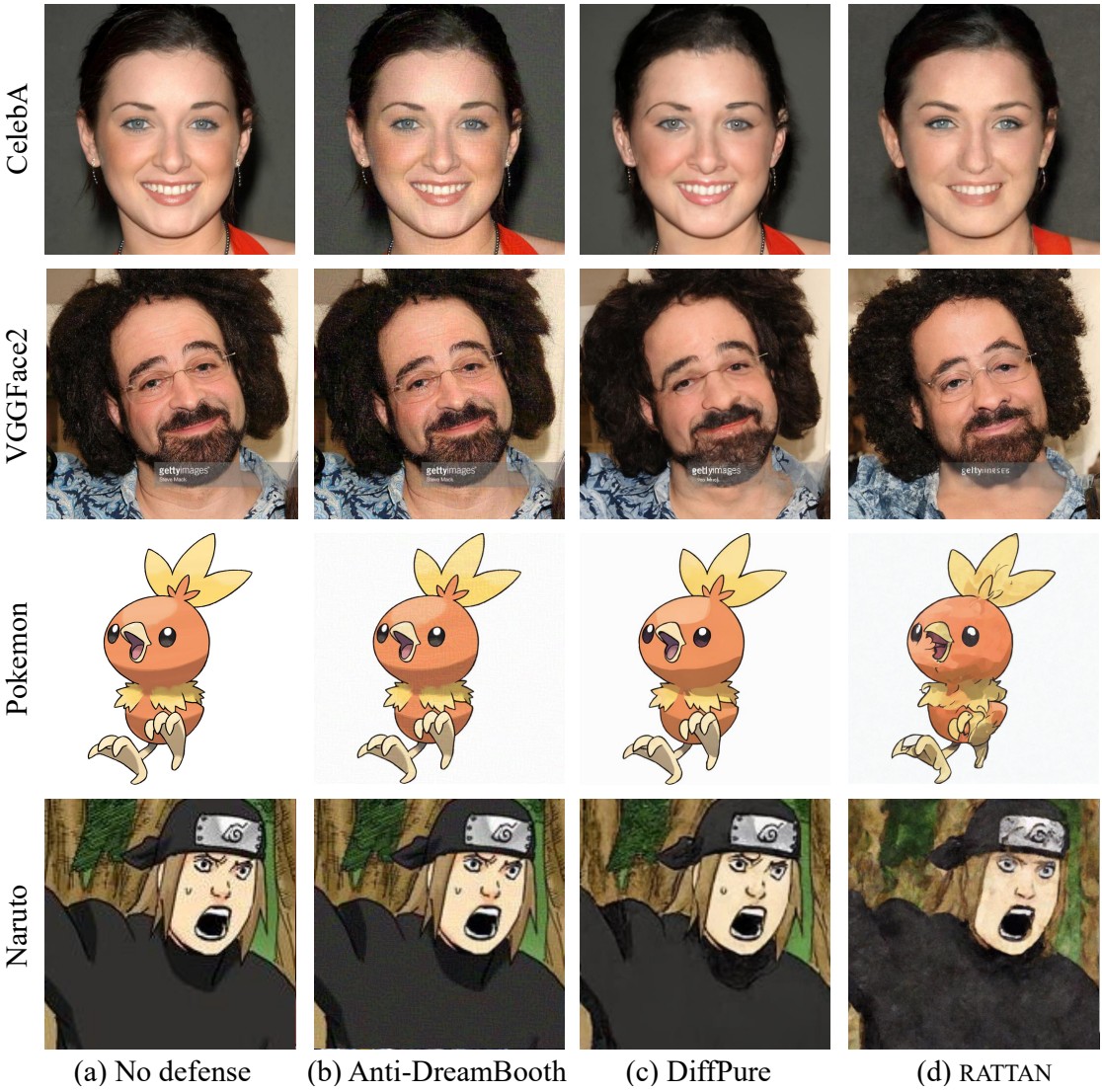

(a) No defense     (b) Anti-DreamBooth     (c) DiffPure     (d) RATTAN

Figure 8: Training dataset samples for Anti-DreamBooth experiments.

relative to (b) and bring the data closer to the no-defense baseline (a). We note that RATTAN can introduce minor stroke/line adjustments in some samples, but these changes do not degrade downstream generation quality and reliably surpass the Anti-DreamBooth protection.

Table 8 summarizes Anti-DreamBooth on non-face domains. On both datasets evaluated, Anti-DreamBooth severely degrades quality. DiffPure lowers local artifacting (BRISQUE) and narrows the distribution gap (FID), but RATTAN closes that gap further with the BRISQUE and FID Scores. BRISQUE is sensitive to local noise/texture statistics, so DiffPure's light denoising can potentially yield a lower BRISQUE score. FID, in contrast, reflects feature-level distribution alignment. RATTAN's text-conditioned re-generation at moderate $\gamma$ value (0.6) overwrites the Anti-DreamBooth coating, pulling samples back toward the clean data manifold and thus consistently improving FID.

Table 8: Evaluation metrics with Anti-DreamBooth poisoned training data for Pokemon and Naruto datasets and cleaning methods before and after applying RATTAN.

| Dataset | Method | BRISQUE ↓ | FID ↓ |
|---------|--------|-----------|-------|
| Pokemon | Original | 22.03 | 210.26 |
|  | Anti-DreamBooth | 39.86 | 336.61 |
|  | DiffPure | 22.04 | 238.38 |
|  | RATTAN | 24.16 | 233.35 |
| Naruto | Original | 24.88 | 276.15 |
|  | Anti-DreamBooth | 35.67 | 387.49 |
|  | DiffPure | 22.68 | 277.17 |
|  | RATTAN | 20.16 | 216.84 |

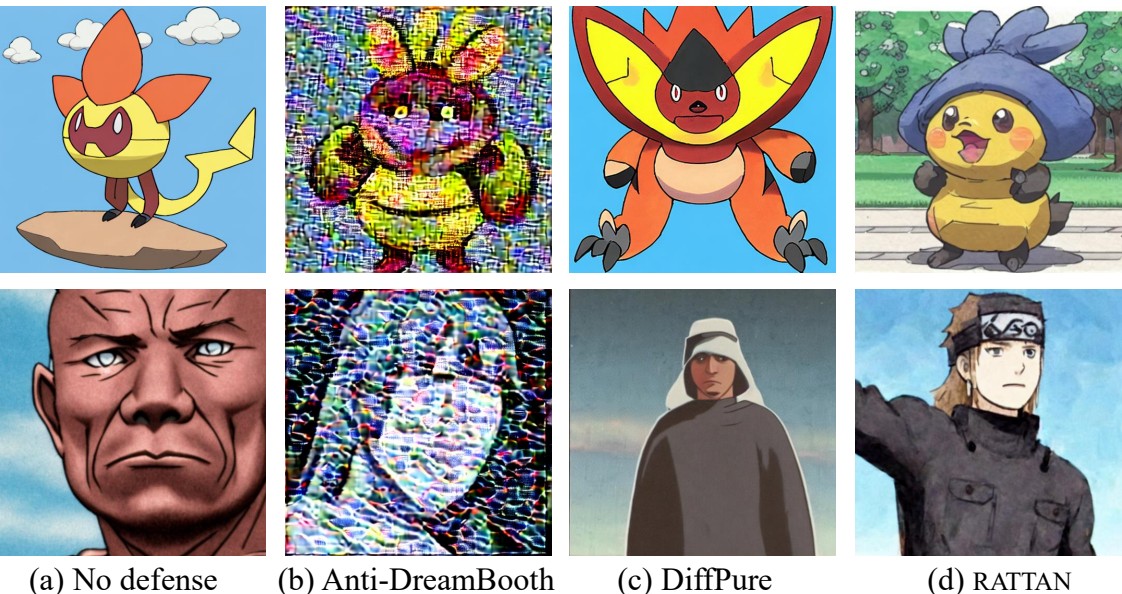

| (a) No defense | (b) Anti-DreamBooth | (c) DiffPure | (d) RATTAN |

Figure 9: Comparison of images generated by different models using the prompt "A photo of a sks pokemon" for the pokemon dataset (top), and "A photo of sks man" for the Naruto dataset (bottom).

Figure 9 shows generations from models fine-tuned on Anti-DreamBooth-poisoned Pokemon and Naruto datasets ((a) no protection, (b) Anti-DreamBooth, (c) DiffPure, (d) RATTAN). We see that Anti-DreamBooth works well at protecting the dataset from mimicry as shown in column (b). Both DiffPure (c) and RATTAN (d) successfully circumvent Anti-DreamBooth and yield images similar to the no-protection baseline (a). However, it is interesting to note that in most of our experimental results, we see a recurring pattern – while DiffPure and RATTAN may both evade such non-trainable defenses, RATTAN allows the fine-tuned models on the more animated datasets such as Pokemon and Naruto to stay more consistent with the original training data than DiffPure. For example, after training on the Naruto dataset and purifying with both methods, we notice that with the prompt "generate a photo of sks man," the models fine-tuned via DreamBooth with DiffPure purified data often generate photo-realistic images, whereas we did not observe out-of-style examples from RATTAN. This observation is consistent with Table 8, where the RATTAN FID scores are lower than DiffPure. Broader validation on additional non-facial datasets is left for future work.

We believe this is a result of DiffPure's purification strategy projecting features of the images towards the base model's natural-image manifold, while RATTAN's text-guided diffusion paradigm lowers the possibility of style drift away from the dataset by anchoring denoising to the target style via text conditioning.

## A.5  DIAGNOSIS Efficacy on WikiArt

We evaluate DIAGNOSIS on the WikiArt dataset to assess its effectiveness in a real-world intellectual property protection scenario involving artist-created works. Unlike previous experiments on structured datasets like Pokemon and CelebA, WikiArt comprises highly diverse artistic styles, making it a strong test case for the robustness of DIAGNOSIS's protection and its subsequent removal by RATTAN. The WikiArt dataset also does not contain captions that fuels RATTAN's text-guided controlled generation process. Hence, we generate captions for each image tested via the BLIP-2 (Li et al., 2023a) model. We used the following system prompt for caption generation:

> "Generate a detailed description of this artwork, capturing its artistic style, color palette, mood, and subject matter. Use expressive and evocative language."

We apply the DIAGNOSIS protection to the artworks and train 10 models on the modified data. Table 9 reports the results of DIAGNOSIS using different coating strengths. Observe that DIAGNOSIS completely fails to detect malicious models, achieving a 0% detection rate for strengths 2 and 3. The detection rate

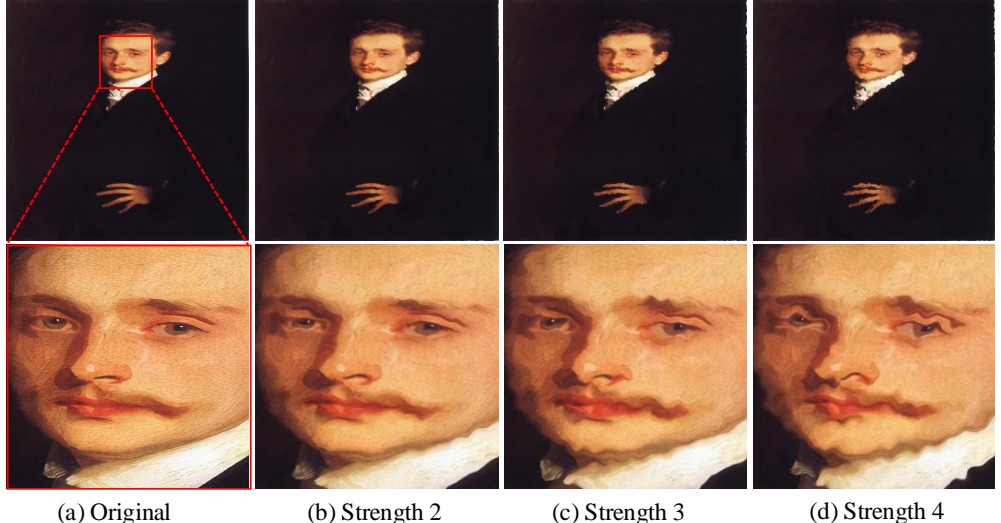

| (a) Original | (b) Strength 2 | (c) Strength 3 | (d) Strength 4 |

Figure 10: ~~Varying DIAGNOSIS coating strengths applied to a WikiArt sample.~~

~~increases slightly at strength 4, reaching 20%, but remains unsatisfactory for detecting unauthorized data usage.~~

Table 9: ~~Results of DIAGNOSIS on WikiArt using different coating strengths.~~

| Strength | Detection | Memorization |
|----------|-----------|--------------|
| 2 | 0% | 0.455 |
| 3 | 0% | 0.462 |
| 4 | 20% | 0.600 |

## A.6 ~~Performance of Image Transformations~~

~~As discussed in Appendix A.1, one straightforward idea to remove the embedded pattern is to apply image transformations. We have shown earlier that Gaussian blur, JPEG compression, and color jittering cannot remove the pattern embedded by DIAGNOSIS. Here, we evaluate six more image transformations, including saturation increase, using 8-bit quantization, adding a green hue, increasing the contrast, cropping by a factor of 1.5 on each side, and increasing the brightness.~~

Table 10: ~~The effect of image transformations on DIAGNOSIS.~~

| Transformation | FID ↓ | Detection ↓ | Memorization ↓ |
|----------------|-------|-------------|----------------|
| Saturation | 229.80 ± 6.15 | 90% | 0.874 |
| 8-bit Quant. | 223.30 ± 7.76 | 90% | 0.860 |
| Hue Shift (Green) | 243.14 ± 8.64 | 100% | 0.896 |
| Contrast | 234.51 ± 7.83 | 70% | 0.718 |
| Cropped | 234.68 ± 7.01 | 100% | 0.852 |
| Brightness | 232.24 ± 7.39 | 90% | 0.840 |

~~As shown in Table 10, we observe that DIAGNOSIS is highly robust against most image transformations. Increasing the contrast in the training set can reduce DIAGNOSIS's effectiveness to some extent but is still limited. The failure of standard image transformations to remove DIAGNOSIS's embedded pattern highlights the need for more sophisticated removal strategies, such as RATTAN.~~

## A.7 Evaluation on Different Models

The experiments in Section 5 of the main text are conducted on Stable Diffusion v1.4. In this section, we evaluate the efficacy of RATTAN against other popular models, including Stable Diffusion v2.0 and Stable Diffusion v2.1. We also report the efficacy of RATTAN against popular rectified-flow models including Qwen-Image (Wu et al., 2025) and FLUX.1-dev (Labs, 2024). We only evaluate the rectified-flow models with DreamBooth fine-tuning since LoRA fine-tuning on these models would require developing a new method built on existing protection techniques, which is out of the scope for this paper. We use a similar setup to the experiemnts on Anti-DreamBooth and StyleGuard from Section 5 to fine-tune these models.

Table 11: Evaluation of RATTAN against DIAGNOSIS on the Pokemon dataset with different Stable Diffusion models.

| Model | Method | Detection ↓ | FID ↓ | Memorization ↓ |
|---|---|---|---|---|
| SD v1.4 | DIAGNOSIS | 100% | 214.86 ± 9.02 | 0.830 |
| | RATTAN | 0% | 211.59 ± 3.15 | 0.327 |
| SD v2.0 | DIAGNOSIS | 100% | 241.12 ± 6.82 | 0.952 |
| | RATTAN | 0% | 242.66 ± 13.96 | 0.444 |
| SD v2.1 | DIAGNOSIS | 90% | 236.01 ± 9.46 | 0.872 |
| | RATTAN | 0% | 240.44 ± 9.06 | 0.378 |

Table 12: Evaluation of RATTAN against DIAGNOSIS on the Pokemon dataset with rectified-flow models.

| Model | Method | Detection ↓ | FID ↓ | Memorization ↓ |
|---|---|---|---|---|
| Qwen-Image-2512 | UNPROTECTED | 0% | 261.26 ± 4.21 | 0.226 |
| | DIAGNOSIS | 40% | 249.80 ± 5.08 | 0.546 |
| | RATTAN | 0% | 259.77 ± 6.26 | 0.172 |
| FLUX.1-dev | UNPROTECTED | 0% | 247.91 ± 6.22 | 0.000 |
| | DIAGNOSIS | 90% | 275.59 ± 7.64 | 0.771 |
| | RATTAN | 0% | 280.22 ± 9.92 | 0.471 |

The results, reported in Table 11, demonstrate that DIAGNOSIS successfully embeds the pattern across all Stable Diffusion models with a detection rate over 95%. However, it is not resilient to RATTAN, which effectively removes the embedded pattern from every model. RATTAN achieves a 100% evasion of detection on the models by DIAGNOSIS, converting all true positives into false negatives while leaving benign models unaffected. Table 12 shows the results of Qwen-Image and FLUX.1-dev after DreamBooth fine-tuning. Here, DIAGNOSIS is able to embed its pattern most successfully in the FLUX model (90% detection rate), while its performance suffers on the larger Qwen-Image model (only 40% detection rate). Again, it is not resilient to RATTAN, which achieves a 100% evasion of detection. This demonstrates the robustness of RATTAN across various model architectures or versions.

## A.8 Visualizations

In this section, we present visualizations of images generated during the controlled generation process of RATTAN, along with visualizations of images produced by fine-tuned text-to-image models.

**Controlled Generation Diffusion Process**

RATTAN utilizes the diffusion process to generate a new image based on the original protected image and its corresponding text. This process involves several steps to progressively denoise the added Gaussian noise. We use 60 steps as the default setting and show the intermediate images produced during this process. The results are illustrated in Table 11 and Figure 12, with $\gamma = 0.6$ and $\gamma = 1.0$, respectively.

With $\gamma = 0.6$, noise is not added to the original protected image until it fully becomes Gaussian noise; instead, the process is stopped at 60% of the noise-adding stage, as discussed in Section 4. From Figure 11(b), it can be observed that the image retains the high-level key features of the protected input shown in (a). The artifacts from the embedded pattern are largely removed by the introduced noise. The subsequent denoising steps gradually refine the image's details, culminating in the final output in (f). The final image retains all the key features of the protected input in (a) but is free from the embedded pattern.

Figure 12 illustrates the intermediate results with a higher $\gamma$ value. The stronger noise significantly obscures the high-level features, as seen in (b). As a result, the subsequent denoising process struggles to retain these features, instead generating an image primarily based on the model's inherent generation capabilities. In the final output, shown in (f), the features are very different from those in (a), and the generated image

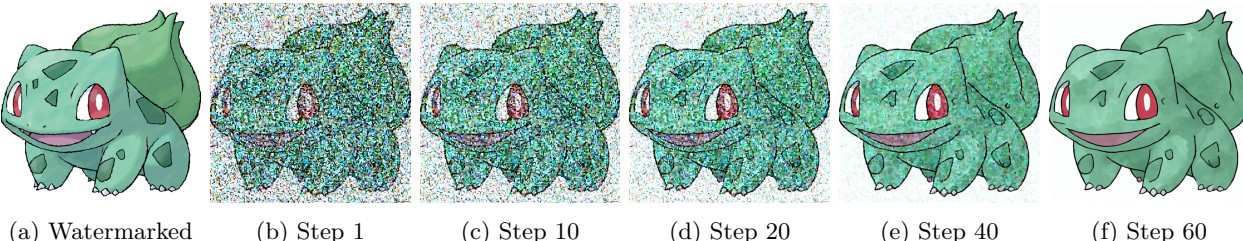

(a) Watermarked     (b) Step 1     (c) Step 10     (d) Step 20     (e) Step 40     (f) Step 60

Figure 11: Intermediate images during controlled image generation of RATTAN with $\gamma = 0.6$.

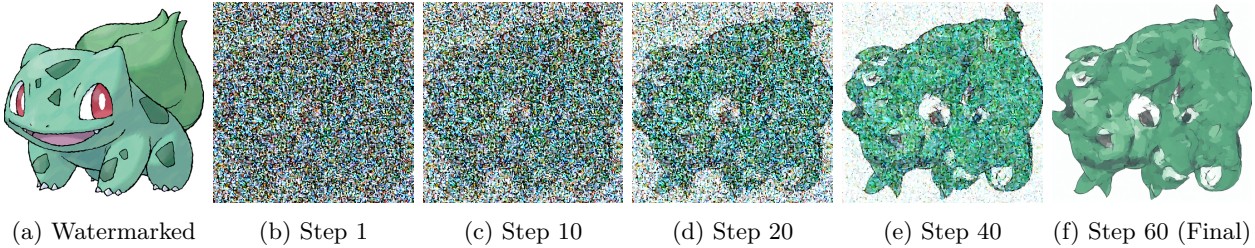

(a) Watermarked     (b) Step 1     (c) Step 10     (d) Step 20     (e) Step 40     (f) Step 60 (Final)

Figure 12: Intermediate images during controlled image generation of RATTAN with $\gamma = 1.0$.

no longer resembles the original input. Therefore, a smaller $\gamma$ is preferred in RATTAN to preserve the main features better.

**Effect of $\gamma$**

In this section, we visually examine the controlled generation results using various $\gamma$ values tested in the ablation study presented in the main paper. Figure 13 displays the images generated with different parameters, demonstrating the influence of $\gamma$ on output quality.

As expected, increasing the $\gamma$ value introduces more noise into the initial input image, leading to a more significant divergence from the original input. This effect is particularly evident in our results, especially in (e) and (f), where the images exhibit significant degradation and divergence from the original protected input image (a). Consequently, models trained with these settings tend to have a higher FID score (indicating lower generated image quality), as reported in Table 4 in the main text.

On the other hand, a smaller $\gamma$ value ensures that the controlled generation closely follows the original input, thereby preserving high-level features. However, this also means that the embedded artifacts are retained, as shown in (b) and (c). Models trained on these generated images can still be detected by the protection method, as also noted in Table 4.

We find that $\gamma = 0.6$ achieves an optimal balance between evading the protection and retaining the key features of the original input image.

**Generated Images by T2I Models**

In this section, we present examples of images generated under three scenarios: a benign model fine-tuned on the Pokemon dataset without any protections, a pattern-embedded model produced by DIAGNOSIS, and a cleansed model by RATTAN.

The results are presented in Figure 14. The first row displays images generated by a benign model. The second and third rows show images generated by DIAGNOSIS-protected models using an unconditional pattern and a trigger-conditioned pattern, respectively. The final two rows depict images generated by cleansed models from RATTAN. The results show minimal visual quality loss in the images between DIAGNOSIS and RATTAN. Most images successfully reproduce similar subjects, retaining key attributes such as colors, creature types, and positioning within the image. This highlights that RATTAN effectively removes the em-

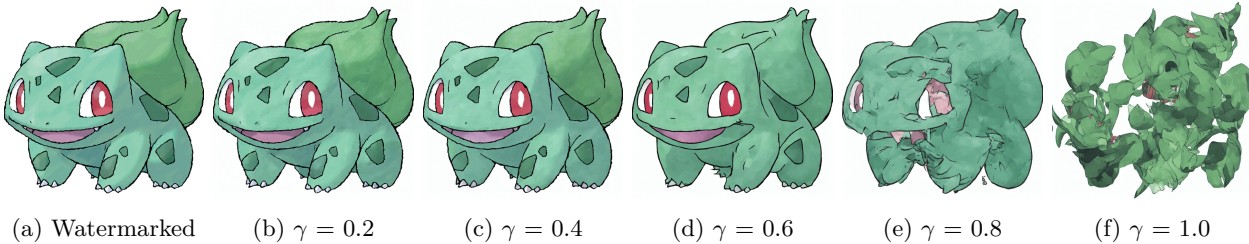

(a) Watermarked     (b) $\gamma = 0.2$     (c) $\gamma = 0.4$     (d) $\gamma = 0.6$     (e) $\gamma = 0.8$     (f) $\gamma = 1.0$

Figure 13: Effect of $\gamma$ on RATTAN's controlled generation process.

Figure 14: Generated images from the benign model, DIAGNOSIS-protected model, and RATTAN-cleansed model. Trig. refers to the use of a trigger-conditioned pattern.

bedded patterns without compromising the generative performance of cleansed models. This ensures that the model's utility remains intact for the adversary, enabling it to generate content in the style of copyrighted material without facing the associated consequences. This underscores the critical need for developing more robust and effective methods to protect intellectual property and private data.

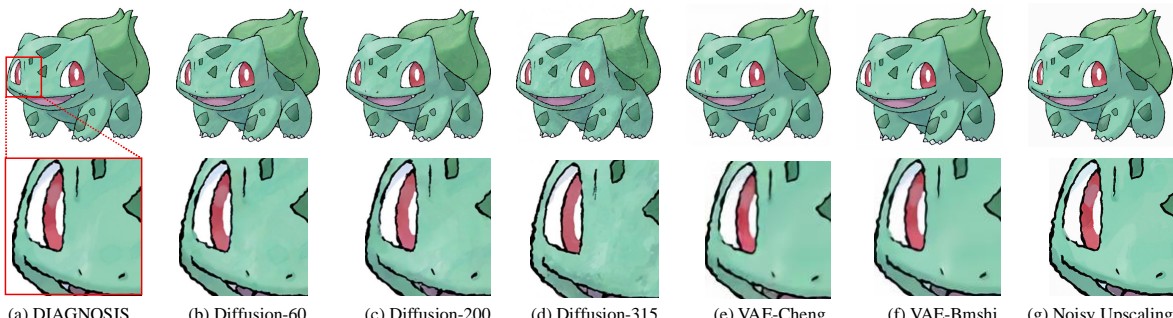

| (a) DIAGNOSIS | (b) Diffusion-60 | (c) Diffusion-200 | (d) Diffusion-315 | (e) VAE-Cheng | (f) VAE-Bmshj | (g) Noisy Upscaling |

Figure 15: Comparison of (a) DIAGNOSIS modified image, and the images processed by baseline methods: Regeneration Attack (Zhao et al., 2024) with 60 noise steps (b), 200 noise steps (c), 315 noise steps (d), VAE-Cheng2020-3 (e), and VAE-Bmshj2018-3 (f). (g) is Noisy Upscaling (Hönig et al., 2025). The bottom row provides a zoomed-in view.

Table 13: Results of Gaussian Noising and Noisy Upscaling against DIAGNOSIS protection.

| Dataset | Method | FID ↓ | Detection ↓ | Memorization ↓ |
|---------|--------|-------|-------------|----------------|
| Pokemon | DIAGNOSIS | 214.86 ± 9.02 | 100% | 0.830 |
| | Gaussian Noising | 240.63 ± 3.64 | 100% | 0.996 |
| | Noisy Upscaling | 234.21 ± 2.24 | 100% | 0.950 |
| Naruto | DIAGNOSIS | 240.13 ± 7.15 | 100% | 0.790 |
| | Gaussian Noising | 235.46 ± 3.55 | 100% | 1.000 |
| | Noisy Upscaling | 226.35 ± 4.53 | 100% | 0.825 |
| CelebA | DIAGNOSIS | 237.63 ± 6.33 | 100% | 0.996 |
| | Gaussian Noising | 235.24 ± 7.15 | 100% | 1.000 |
| | Noisy Upscaling | 224.54 ± 1.99 | 100% | 0.860 |

## A.9 Results of Baselines on DIAGNOSIS

We evaluate Noisy Upscaling by Hönig et al. (2025) and Regeneration Attack by Zhao et al. (2024) using various parameters and configurations to assess their efficacy in removing the memorization of protected patterns by T2I models.

**Noisy Upscaling.** While Hönig et al. (2025) propose several methods, we adopt Noisy Upscaling for the main evaluations in this work, as it provides the best performance according to their paper. Here, we evaluate another method, Gaussian Noising, against the protection by DIAGNOSIS.

The results in Table 13 show that both techniques proposed by Hönig et al. (2025) fail to eliminate the embedded pattern. In the case of the Pokemon and Naruto datasets, the attacks even amplify the memorization strength and FID scores. Figure 15 (g) shows the purified image from Noisy Upscaling. We can still observe the warped coating by DIAGNOSIS. Hence, these techniques are limited against semantic features leveraged by protections such as DIAGNOSIS.

**Regeneration Attack.** Regeneration Attack modifies the latent representation of an input image by adding random noise, and then performs the diffusion denoising process to regenerate a modified version of the input. The *noise step* parameter in Regeneration Attack controls the level of perturbation. More noise steps introduce greater levels of noise, pushing the latent representation further away from its original state. However, unlike the image-to-image diffusion process, Regeneration Attack follows a variance-preserving stochastic differential equation (VP-SDE), which maintains the variance of the latent distribution. This allows certain structural features to persist, including structural distortions introduced by DIAGNOSIS.

Table 14 reports the results of Regeneration Attack using different parameters and configurations. We test different noise steps: 60, 200 (maximum noise steps tested in the paper), and 315. The results in the table show that none of these parameters can successfully evade DIAGNOSIS. We also evaluate different reconstruction models used in Regeneration Attack such as VAE-Bmshj2018-3 and VAE-Cheng2020-3. The results are similar. Figure 15 displays the processed images by Regeneration Attack from the DIAGNOSIS-protected input. Observe that the warping effect persists. Increasing the noise step, particularly to 315,

Table 14: Results of Regeneration Attack with different configurations against DIAGNOSIS protection.

| Dataset | Method | FID ↓ | Detection ↓ | Memorization ↓ |
|---------|--------|-------|-------------|----------------|
| Pokemon | DIAGNOSIS | 214.86 ± 9.02 | 100% | 0.830 |
| | Diffusion-60 | 249.27 ± 4.25 | 100% | 0.982 |
| | Diffusion-200 | 230.87 ± 2.45 | 100% | 0.940 |
| | Diffusion-315 | 233.76 ± 6.12 | 100% | 0.985 |
| | VAE-Bmshj2018-3 | 260.25 ± 8.92 | 100% | 0.975 |
| | VAE-Cheng2020-3 | 258.78 ± 7.00 | 100% | 0.965 |
| Naruto | DIAGNOSIS | 240.13 ± 7.15 | 100% | 0.790 |
| | Diffusion-60 | 241.79 ± 4.02 | 100% | 0.939 |
| | Diffusion-200 | 241.23 ± 5.24 | 100% | 0.958 |
| | Diffusion-315 | 250.38 ± 2.45 | 100% | 0.965 |
| | VAE-Bmshj2018-3 | 268.72 ± 6.22 | 100% | 0.985 |
| | VAE-Cheng2020-3 | 257.48 ± 5.88 | 100% | 0.985 |
| CelebA | DIAGNOSIS | 237.63 ± 6.33 | 100% | 0.996 |
| | Diffusion-60 | 237.20 ± 4.66 | 100% | 0.965 |
| | Diffusion-200 | 240.20 ± 2.37 | 100% | 0.947 |
| | Diffusion-315 | 241.01 ± 6.68 | 100% | 0.967 |
| | VAE-Bmshj2018-3 | 256.76 ± 5.72 | 100% | 0.955 |
| | VAE-Cheng2020-3 | 259.79 ± 4.99 | 100% | 0.961 |

alters the texture of the objects but preserves high-level features, such as the DIAGNOSIS-coated lines, demonstrating that these artifacts remain robust to Regeneration Attack.

## A.10 Hyper-Parameters in Experiments

In this section, we present the hyper-parameters used for fine-tuning with each algorithm to aid reproducability. These hyper-parameters and setup are also included in the publicly-released code.

Note that while training multiple models, the seeds are incremented from 1-10 during LoRA training.

**DIAGNOSIS (LoRA fine-tuning).** We fine-tune Stable Diffusion v1.4 with LoRA at 512×512 resolution using random horizontal flip and captions from `additional_feature`. Optimization uses AdamW with a constant learning rate of $1 \times 10^{-4}$ (no warmup), batch size 1, and 100 epochs. Training runs in FP16 mixed precision with seed 42.

**RATTAN (LoRA fine-tuning).** We apply the same setup as above (Stable Diffusion v1.4, $512 \times 512$, random flip, captions from `additional_feature`, AdamW, constant scheduler, no warmup, FP16, seed 42, W&B reporting), but use a learning rate of $1 \times 10^{-5}$, batch size 1, 30 epochs, and checkpoint every 2,000 steps.

**Anti-DreamBooth ASPL.** We train Stable Diffusion 2.1-base with xFormers memory-efficient attention, prior preservation (`prior_loss_weight`=1.0), and text-encoder training at $512 \times 512$ with center crop. Optimization uses a constant learning rate of $5 \times 10^{-7}$, batch size 1, and 50 training steps; inner loops use `max_f_train_steps`= 3 and `max_adv_train_steps`= 6. We apply PGD with step size `pgd_alpha`= $5 \times 10^{-3}$ and radius `pgd_eps`= $5 \times 10^{-2}$, with checkpointing every 10 iterations.

**DreamBooth (after ASPL).** We fine-tune Stable Diffusion 2.1-base at $512 \times 512$ with center crop, prior preservation (`prior_loss_weight`=1.0), and text-encoder training. Optimization uses a constant learning rate of $5 \times 10^{-7}$ (no warmup), batch size 2 with gradient accumulation 1, and 1000 training steps; checkpointing every 500 steps. We use `bf16` mixed precision (including prior-generation precision) and `sample_batch_size`= 8.

**RATTAN** employs the same DreamBooth hyper-parameters; only the training data differs (controlled-generated instead of perturbed).

### A.11 LLM Usage Disclosure

We used an LLM to assist with minor grammatical, stylistic, and typographical corrections during paper preparation. The models were not used for generating ideas, experiments, analyses, or substantive writing.

