# OpenReview forum: "Evading Protections Against Unauthorized Data Usage via Limited Fine-tuning"
_TMLR — Rejected by TMLR_

### Review · Reviewer_Ld2y · 2026-02-21

**Summary Of Contributions:**

This paper proposes Rattan, a method that removes protective perturbations from images via a controlled image generation scheme (partial forward noising that preserves only the high-level structure of protected images, combined with an unmarked Stable Diffusion model). It then fine-tunes the marked model to induce forgetting of the learned protected features, thereby mitigating risks of intellectual property infringement or privacy leakage in image generation. Evaluations on four datasets and 300 text-to-image diffusion models show that Rattan can reduce the detection accuracy of existing strong protection methods to 0%.

**Audience:**

Yes

**Audience Explanation:**

This paper addresses privacy concerns in image generation and proposes a two-stage approach: it uses an unmarked Stable Diffusion model to construct samples that preserve only high-level information, and then fine-tunes a marked Stable Diffusion model to induce forgetting of the learned signature. The topic is well aligned with the scope of TMLR.

**Claims And Evidence:**

No

**Claims Explanation:**

1. The paper does not provide a rigorous guarantee that the proposed controlled image generation scheme will consistently remove sensitive patterns. Key hyperparameters $\gamma$ is chosen empirically. As a result, their effectiveness may not generalize when the underlying diffusion backbone or the data distribution changes.

2. The approach seems primarily tailored to removing low-level, pattern-like artifacts. It is unclear how the method would apply when the protected signal is inherently high-level or semantic (e.g., a distinctive color palette or stylistic characteristics), which may be precisely the attributes one aims to protect.

3. The experimental evaluation is largely restricted to Stable Diffusion. Additional evidence across other text-to-image diffusion architectures would strengthen the claims about generality.

4. Based on the qualitative results (e.g., Fig. 11 and Fig. 5), the Rattan-generated images sometimes exhibit noticeable quality degradation. For instance, in Fig. 5 (Naruto), the regenerated outputs appear less detailed and less visually faithful than the originals.

**Requested Changes:**

See my question above.

---

> ### Author Response · Authors · 2026-04-03
> **Official Comment by Authors**
>
> We thank the reviewer for the detailed review. We address each concern below. In the revision, new additions are in blue font, and deleted text is in orange.
>
> ## No rigorous guarantee
> We agree that RATTAN does not provide a formal guarantee of pattern removal in a mathematical proof sense. The paper frames the contribution as an empirical attack demonstrating that current protections have concrete, exploitable failure modes, and our evaluation across four datasets, four protection methods, 300 models, and two model architectures (including the newly added Qwen-Image and Flux.1-dev) provides strong empirical evidence. The parameter γ is selected via ablation (Table 6) and is shown to be robust across a range of values (0.4-0.8).
>
> ## Applicability to high-level or semantic signals
> According to a user study of 1,000 artists conducted by existing work [1], artists overwhelmingly prefer invisible, low-disruption watermarking approaches over visible or style-altering ones, confirming that the threat model we study is practical. We have clarified this need for a low-level and imperceptible attack supported by the survey in our threat model (Section 3) in the revised version.
>
> _[1] Shan, Shawn, et al. "Glaze: Protecting artists from style mimicry by {Text-to-Image} models." 32nd USENIX Security Symposium (USENIX Security 23). 2023._
>
> ## Evaluation restricted to Stable Diffusion
> We have included new experimental results with other model architectures for controlled generation in the revision. Table 6 shows results for newly added Qwen-Image, and FLUX.1-dev. These two are flow-matching models with a fundamentally different generative architecture from Stable Diffusion. RATTAN can reduce the detection rates using these models to 0%. Due to time constraints in applying DIAGNOSIS to these different T2I model architectures as they are much larger (12B FLUX.1-dev vs. ~1B Stable Diffusion), we are still in the process of training the protected models. We will update the results once they are available.
>
> ## Image quality degradation in RATTAN generated images
> We thank the reviewer for the comment. The images shown in column (f) of Fig. 5 (Fig. 8 in the revised version) are intermediate outputs used solely for fine-tuning the marked model, as RATTAN aims to strip the embedded pattern before regeneration. These intermediate images are not the final output. Table 1 shows that RATTAN maintains or improves FID relative to DIAGNOSIS-embedded models across all three datasets and both attack types, and Table 3 shows RATTAN recovering BRISQUE and FID to near-original levels against Anti-DreamBooth.
> Fig. 11 (Fig. 14 in the revised version) shows final model-generated outputs on the Pokemon dataset. We believe the RATTAN-trained model produces images that are frequently comparable to the benign baseline, and often better than the DIAGNOSIS-trained model. We acknowledge that visual quality judgments are inherently subjective, which is why we prioritize objective quantitative measures such as FID and BRISQUE, and RATTAN can effectively preserve as shown in Table 3.

---

> > ### Author Response · Authors · 2026-04-10
> > **Official Update by Authors**
> >
> > ## Update on Rectified Flow Model Experiments
> > We have completed training the DIAGNOSIS-protected models for Qwen-Image and FLUX.1-dev and have updated the results in the revision (Table 12, Appendix 7). DIAGNOSIS achieves a 90% detection rate on FLUX.1-dev but struggles on the larger Qwen-Image model with only a 40% detection rate. Importantly, RATTAN successfully evades DIAGNOSIS detection in both cases (0% detection rate), consistent with our results on Stable Diffusion v1.4, v2.0, and v2.1. These results reinforce that RATTAN generalizes across differentT2I architectures.

---

### Review · Reviewer_w1mc · 2026-03-05

**Summary Of Contributions:**

Watermarking methods have been developed to prevent fine-tuning or personalization using someone’s existing data. These approaches can be broadly divided into two categories. DIAGNOSIS embeds imperceptible patterns into images and later checks whether those patterns can be detected in a trained model. Anti-Dreambooth implants adversarial perturbations into watermarked images so that training on them severely degrades the model.

This paper introduces a method called RATTAN, which demonstrates that both types of watermarking approaches can be bypassed or removed with only a few rounds of fine-tuning. RATTAN works by regenerating images to remove the subtle watermark patterns, producing images that remain visually similar to the originals but no longer contain the fine-grained signals. The model is then re-trained on these cleaned images, causing it to forget the embedded patterns.

Across diverse datasets and benchmarks, the paper shows that RATTAN can reduce detection rates of systems such as DIAGNOSIS to 0% and recover the quality degradation caused by adversarial watermarking (Anti-DreamBooth).

**Audience:**

Yes

**Audience Explanation:**

Watermarking for Text-to-Image model is important and timely area. There should be some TMLR's audience that would interested in this work.

**Claims And Evidence:**

Yes

**Claims Explanation:**

The paper conducts experiments on the DIAGNOSIS method using three datasets (Pokemon, Naruto, CelebA), and on Anti-DreamBooth using two datasets (CelebA, VGGFace2). It also includes comparisons with image purification methods such as DiffPure.

DIAGNOSIS shows clear performance improvements, while Anti-DreamBooth achieves results that are comparable to or slightly worse than DiffPure. However, the overall evidence is solid and well-supported.

That said, conducting experiments only on Stable Diffusion is a clear limitation.

**Requested Changes:**

Optional, but strongly recommend doing experiments on state-of-the-art T2I model, e.g., FLUX or Qwen-Image.

---

> ### Author Response · Authors · 2026-04-03
> **Official Comment by Authors**
>
> We thank the reviewer for the positive assessment and the constructive suggestion. In the revision, new additions are in blue font, and deleted text is in orange.
>
> ## Experiments on SOTA T2I models
> We add two SOTA T2I models for controlled generation during rebuttal. Table 6 includes results for both Qwen-Image and Flux.1-dev as backbones. Both detection rates are reduced to 0% with memorization strengths of 0.060 and 0.129 respectively, and FID scores comparable to SD v1.4. Due to time constraints in applying DIAGNOSIS to these different T2I model architectures as they are much larger (12B FLUX.1-dev vs. ~1B Stable Diffusion), we are still in the process of training the protected models. We will update the results once they are available.

---

> ### Author Response · Authors · 2026-04-10
> **Official Update by Authors**
>
> ## Update on Rectified Flow Model Experiments
> We have completed the DIAGNOSIS-protected training for Qwen-Image and FLUX.1-dev and updated the results accordingly. The full experimental details and results table (Table 12) are provided in Appendix 7. DIAGNOSIS achieves a 90% detection rate on FLUX.1-dev but only 40% on the larger Qwen-Image model. Notably, RATTAN evades DIAGNOSIS detection entirely (0%) on both architectures, consistent with our findings on Stable Diffusion models. These results further support the generalizability of RATTAN across both UNet-based and rectified flow T2I models.

---

### Review · Reviewer_WoyG · 2026-03-19

**Summary Of Contributions:**

Due to the large volume of review requests I have received, I have to skip certain minor details and focus on the key issues.

**Additional Comments:**

Due to the large volume of review requests I have received, I have to skip certain minor details and focus on the key issues that the authors need to address.

**Audience:**

Yes

**Audience Explanation:**

As TMLR allows the submission from different fields, the people in the safety community can feel interested to this work.

**Broader Impact Concerns:**

This work is actually to attack the existing safety system. Though it can raise people's awareness to the potential safety issues, the authors may still consider introducing some potential defense strategy.

**Claims And Evidence:**

Yes

**Claims Explanation:**

This work has conducted necessary experiments to support its methods.

**Requested Changes:**

1. My primary concern is that this manuscript is being submitted to a journal rather than a conference. For journal publications, key and useful experimental results should be presented more prominently in the main body of the paper. Placing substantial portions of them in the appendix is therefore suboptimal and may even create an impression of deliberately concealing important details.

2. My second concern is that the protection methods compared in this paper are quite old. Most of them, such as DIAGNOSIS and Anti-DreamBooth, are from 2023 or earlier. It would be better to also compare with newer protection methods from 2024 or 2025.

3. The paper mainly tests on a few fixed and relatively simple datasets, such as Pokemon, Naruto, and CelebA. These datasets have clear styles and limited diversity. This makes me suspect that the method is specially designed for these datasets and may not generalize well to more complex real-world data.

4. It relies heavily on text prompts for controlled generation and on image-text pairs for fine-tuning. Does this approach work exclusively for T2I tasks? Have the authors considered whether it can be generalized to other modalities, such as text-to-video or unconditional diffusion models?

---

> ### Author Response · Authors · 2026-04-03
> **Official Comment by Authors**
>
> We thank the reviewer for the thoughtful feedback. We address each concern below. In the revision, new additions are in blue font, and deleted text is in orange.
>
> ## Appendix placement of results
> We thank the reviewer for the suggestion. We have revised the manuscript to move the DIAGNOSIS results on WikiArt, as well as the ablation on common image transformations to remove protections into the main body of the paper. We have also included images generated by the fine-tuned models to show the quality of images, with additional examples remaining in the appendix to save space. We have restructured Section 5 to accommodate these without exceeding the 12 page limit.
>
> ## Comparison with newer protection methods
> In the revision, we included StyleGuard (2025) as an additional baseline protection method. It is a state-of-the-art protection approach that employs semantic-level style perturbations, and we believe it represents a stronger and more recent learning-prevention defense than Anti-DreamBooth. Our results show that RATTAN is also effective against StyleGuard. We have included these results on page 10 in the revised main body.
>
> ## Dataset diversity
> We acknowledge that Pokemon, Naruto, and CelebA are relatively constrained domains. However, we want to highlight that this concern partially applies to DIAGNOSIS and coating-based protection mechanisms itself where the embedded pattern relies on a learnable function (in contrast to unlearnable features in methods like Anti-DreamBooth and StyleGuard). Our Appendix A.5 already shows that on WikiArt, DIAGNOSIS fails entirely to achieve reliable detection even without RATTAN, achieving 0% detection rate at coating strengths 2 and 3. This suggests the generalization limitation lies primarily with the protection methods rather than RATTAN. We have also shown that RATTAN is effective against StyleGuard on WikiArt on page 10.
>
> ## Generalization beyond T2I
> Testing such methods on other modalities is an important direction, and we agree they are worth investigating in future work. That said, we note that our ablation in Section 5.3 already demonstrates that RATTAN generalizes beyond traditional DDPM-style diffusion models to flow-matching architectures, specifically Qwen-Image and FLUX.1-dev (Table 6). These models use a fundamentally different generative mechanism (rectified flow / flow matching) yet RATTAN reduces the detection rate to 0% and induces low memorization on both, demonstrating that RATTAN is architecture-agnostic at least within the image domain.
>
> ## Broader impact and defense strategies
> We agree that briefly discussing potential mitigations strengthens the paper's contribution. In the revision, we have expanded Section 7 to discuss directions for more robust defenses, such as stronger unlearnable perturbation methods, which may prove more resilient against purification-based attacks like RATTAN.

---

### Decision · Action_Editor_QiXk · 2026-05-19

**Recommendation:** Reject

**Audience:**

Yes

**Audience Explanation:**

See box above.

**Claims And Evidence:**

No

**Claims Explanation:**

The reviewers and I agree that the submission addresses an important and relevant problem for the TMLR audience, but there are concerns about whether the current evidence is strong enough to support the breadth of the paper’s claims. In particular, the experiments would benefit from broader evaluation beyond Stable Diffusion. Furthermore, the evaluation covers a limited set of protection methods, and the datasets used are relatively simple and stylistically constrained, making it difficult to assess generalization to more complex real world settings.

Some claims appear insufficiently supported: the method depends on empirically chosen hyperparameters, may be mainly suited to removing low level perturbation patterns, and has unclear effectiveness when the protected signal is semantic or high level, such as color palette or style. The evaluation is largely centered on Stable Diffusion, and some qualitative examples show noticeable degradation in regenerated images. Although the authors added additional experiments and responses, the paper would need a more comprehensive empirical validation and a clearer account of its limitations before acceptance.

**Resubmission Of Major Revision:**

The authors may consider submitting a major revision at a later time.